# Decoding phantom limb movements from intraneural recordings

Cecilia Rossi [1], Marko Bumbasirevic[2], Paul Čvančara [3], Thomas Stieglitz [3], Stanisa Raspopovic [4,5], Elisa Donati [1,7] ✉ & Giacomo Valle [6,7] ✉

Limb loss causes severe sensorimotor deficits and often necessitates prosthetic devices, particularly in lower-limb amputees. Although direct neural recording from residual nerves offers a biomimetic route for prosthetic control, low signal amplitudes and challenges in nerve interfacing have limited adoption. Intraneural multichannel electrodes provide a potential solution by enabling access to motor signals from muscles lost after amputation. Here, we report intraneural recordings from two transfemoral amputees using transversal intrafascicular multichannel electrodes implanted in distal branches of the sciatic nerve. We identified multiunit activity associated with volitional phantom movements of the knee, ankle, and toes, exhibiting joint- and direction-specific modulation distributed across electrodes. A Spiking Neural Network–based decoder outperformed conventional methods in predicting attempted movements, with further gains achieved by integrating intraneural and intermuscular signals. Motor and sensory maps showed minimal overlap, indicating early segregation within the sciatic nerve. These findings pave the way for bidirectional, neurally-controlled prosthetic systems.

The human sensorimotor system relies on a tight interplay between voluntary motor commands and continuous sensory feedback to generate coordinated, dexterous movements. Limb loss modifies the brain-body communication, leading to severe sensorimotor disabilities and psychological distress. As a result, many amputees remain wheelchair bound, immobile, or only partially integrated in the activities of daily life. Lower-limb amputations are more prevalent than upper-limb ones, accounting for 69% of all the individuals living with the loss of a limb[1]. Among these, more than half are major amputation[2], i.e., above the ankle level. In addition, higher level of amputation is associated with increased disability, characterized by reduced walking speed and higher energy expenditure (i.e., above-knee amputation (AKA) or hip-disarticulation)[3].

Lower-limb prosthetic devices are commonly used to restore mobility, but their functionality remains limited, particularly in terms of intuitive volitional control and somatosensory feedback integration[4]. Current commercial solutions typically rely on passive mechanisms of action, avoiding direct control by the user[5]. In some cases, surface electromyographic signals (sEMG) could be used to directly control the prosthetic device (i.e., myoelectric prostheses)[6]. However, for lower-limb amputees, these solutions lack the specificity and robustness needed for fine-grained, volitional control, particularly when multiple degrees of freedom (DoF) are involved. A major barrier to the efficacy of sEMG decoders is the challenge of accessing the appropriate muscles. Muscles located deep within the thigh, those that have been anatomically reorganized post-amputation, and those lost due to the amputation, present significant challenges for capturing

[1]Institute of Neuroinformatics, University of Zurich and ETH Zurich, Zurich, Switzerland. [2]Orthopaedic Surgery Department, School of Medicine, University of Belgrade, Belgrade, Serbia. [3]Department of Microsystems Engineering–IMTEK, IMBIT // NeuroProbes, BrainLinks-BrainTools Center, Bernstein Center Freiburg, University of Freiburg, Freiburg, Germany. [4]Department of Health Sciences and Technology, Institute for Robotics and Intelligent Systems, ETH Zürich, Zürich, Switzerland. [5]Center for Medical Physics and Biomedical Engineering, Medical University of Vienna, Vienna, Austria. [6]Department of Electrical Engineering, Chalmers University of Technology, Gothenburg, Sweden. [7]These authors contributed equally: Elisa Donati, Giacomo Valle. ✉e-mail: elisa@ini.uzh.ch; valleg@chalmers.se

reliable movement-related signals through surface recordings[7]. Notably, intramuscular EMG microelectrode arrays have been shown to target deeper muscles with high precision, and they can be used for interfacing online[8,9]. To improve human-machine interfacing[10] of intramuscular EMG and the limitation of noninvasive techniques, novel surgical approaches involving nerve or muscle transfer have been developed and successfully tested for decoding attempted phantom movements[11]: Regenerative peripheral nerve interfaces (RPNIs)[12] and target muscles reinnervation (TMR)[13]. While these approaches have shown promise, they have primarily been evaluated for motor decoding in upper-limb amputees[12,14] or a single above-knee amputation with a few DoF[15]. Agonist-antagonist myoneural interfaces (AMI) have been successfully used in multiple transtibial amputees, significantly improving their gait[16,17]. However, only two studies have reported AMI in pilot trials with above-knee amputees, with promising results[18,19], though further testing is still necessary to validate the approach. While these approaches have successfully decoded movements of the knee or ankle, none have demonstrated the ability to decode movements from all the leg joints, including the foot and toes.

In addition to motor control, somatosensory feedback from the leg and foot is essential for providing meaningful information to the users' brain, supporting sensorimotor control[20], posture, and balance[21]. Recent studies have shown the use of neural interfaces implanted in the tibial, peroneal, and sciatic nerves after limb loss for restoring rich and selective somatosensory information. Results showed multiple functional and cognitive benefits in both transtibial[22,23] and transfemoral amputees[24–28], highlighting the importance of restoring sensory feedback alongside motor control. The integration of the somatosensory feedback with the motor decoding system could further enhance the control and usability of prostheses, creating a bidirectional communication system that mimics natural limb function. Although the implantable electrodes are in close contact with, not only afferent, but also efferent fibers encoding information about the volitional limb movement, none of these electrodes has been used to record electroneurographic (ENG) signals for motor decoding. This is primarily because ENG signals are low in amplitude and highly prone to noise, requiring penetrating electrodes, such as Utah arrays[29] or microneurography needles[30], to capture them reliably and advanced real-time decoders, which have so far hindered the development of neurally-controlled prostheses. Few attempts only in upper-limb amputees, implanted in the median and ulnar nerves, have shown promising off-line decoding of fingers and hand movements[29,31–33]. To date, no studies have attempted to record neural signals and decode leg movements in humans. The ability to directly record and decode the neural activity from peripheral nerves could provide access to a richer set of control signals for prosthetics, including those associated with muscles lost after amputation, enabling faster and more complex movements.

To this aim, we implanted multiple TIMEs in two individuals with above-knee amputation in their distal branch of the sciatic nerve (Fig. 1A, B). We asked the participants to move their phantom leg, including knee, ankle and toes, in different directions (Fig. 1C), while simultaneously recording the intraneural activity (Fig. 1D). Unlike prior studies where TIMEs were used solely for stimulation, we employ them here bidirectionally, capturing phantom movement execution and delivering sensory feedback through the same intraneural sites in above-knee amputees. We used this dataset to conduct a comprehensive analysis of motor decoding and sensory mapping. Firstly, we comprehensively characterized the neural efferent response in different movement conditions (e.g., joint and direction selectivity), including both the neural dynamics and the spatial activation of the 56 implanted active sites (AS) within the nerve. Secondly, we designed and implemented a purposely designed intraneural motor decoder, based on a spiking neural network. The SNN, aligned with the event-driven structure of ENG signals, significantly outperformed standard

classifiers in both accuracy and efficiency in both participants[34]. In addition, we also demonstrated the possibility to use this implantable technology to record inter-muscular activity. Finally, we compared motor maps derived from the intraneural recordings with the sensory maps obtained delivering electrical neurostimulation through the same electrodes.

The use of SNNs is particularly advantageous for decoding ENG signals, as these networks are designed to process the spiking nature of the data, making them a natural fit for this task. Raw ENG spikes, however, are not sufficient for robust decoding. To improve the signal quality for motor control, we introduce an event-based encoding technique, which converts continuous ENG signals into dense spike trains. This encoding method enhances the signal's usability by increasing the density of information available for decoding, providing a more accurate and robust system for prosthetic control.

The use of implantable neurotechnology and neuromorphic hardware[35] not only for neurostimulation, but also for direct peripheral nerve decoding, provides preliminary validation of motor decoding feasibility for bidirectional, neurally-controlled prosthetic limbs.

## Results

We recorded residual neural activity from a total of 56 active channels, implanted inside the distal branch of the sciatic nerve stump in two above-knee amputees (Table S1), who had been asked to perform three different types of phantom motor tasks (ankle movement, knee movement and toes movements) in two different directions (flexion and extension) each (Fig. 1A–C). The tibial branch of the sciatic nerve is responsible for the innervation of the majority of flexor muscles in the lower part of the leg, in particular muscles like the Gastrocnemius, which is accountable for movements of the ankle and knee, as well as the Flexor Digitorum and Flexor Hallucis, instead involved in the flexion of ankle and toes (Fig. 1B). For this reason, we expected to being able to extract neural modulation concurrently with volitional movements of all the three anatomical regions analyzed, as well as being able to decode the direction of the intended phantom motion.

After pre-processing the neural signal, we characterized in detail the neural activity in response to different phantom movements involving multiple joints and directions in both subjects. This neural activation has been then compared to the muscular innervation of the targeted nerve and the specific movement. Then, we designed and implemented neural decoders in order to predict the subjects' intentions to move their phantom limb. We developed SNN-based decoders using the recorded spiking signals as direct inputs, and we compared their performance with that of conventional non-spiking-based decoders. Finally, we expanded the bandwidth of the neural signal to include also the recorded activity of the thigh muscles using the neural interfaces as inter-muscular electrodes. We evaluated the benefits of using hybrid decoders compared to ENG-only decoders.

### The intraneural signal encodes joint-related movements of the lower-limb

From the recorded intraneural activity, we observed significant neural modulation associated with movements of the lower-limb (Fig. 2A). In fact, 91% (51/56 implanted electrodes) of the recording sites showed responsiveness for at least one of the phantom movements analyzed in S1 (Fig. 2C), with only 5 overall silent electrodes, all located on the same TIME. Of these 51 channels, 90% (46/51 electrodes) showed extension-related neural activation, while 71% (36/51 electrodes) resulted associated with flexion-specific modulation (Fig. 2C). For S2, instead, the situation seems to be reverted, with only 37.5% (21/56 electrodes) revealing significant neural modulation during the execution of the ankle and toes phantom motions (no data for knee movements is available for S2) (Supplementary Fig. 2A). Of the active 21/56 electrodes, the 85.7% (18/21 electrodes) shows activation during the flexion

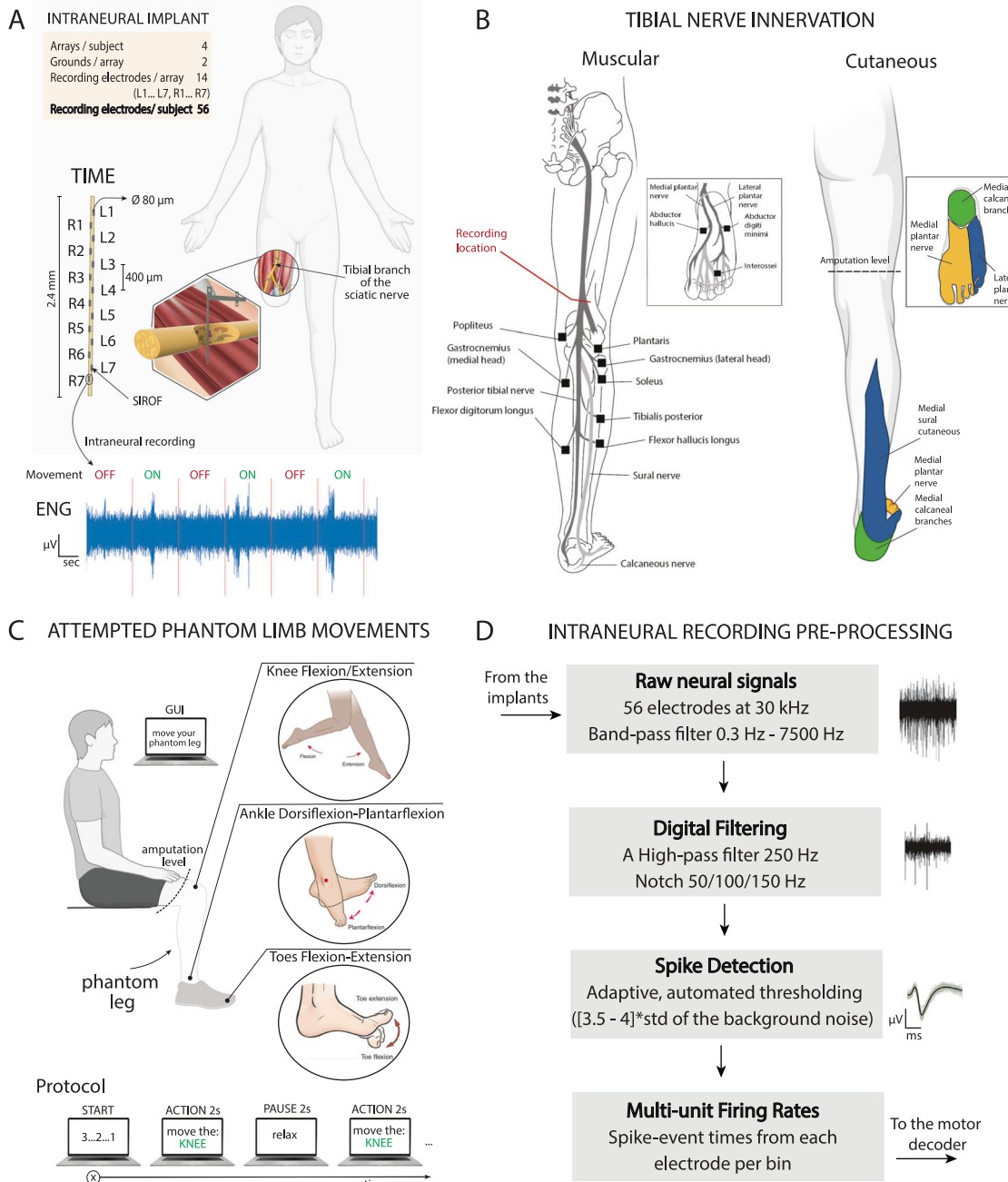

**Fig. 1 | Intraneural recording of the tibial nerve during phantom movements.**
**A** Four intraneural multi-channel electrodes (TIMEs) were implanted in the distal branch of the sciatic nerve of two individuals with transfemoral amputation. Each TIME had 14 active sites for recording and stimulation. **B** Motor and sensory innervations of the tibial nerve. Implants' location shown in red. **C** Subjects were asked to move their phantom limb following the instructions displayed on a screen. The protocol consists in knee-ankle-foot movements of 2 s interleaved with 2 s of pause. **D** Pre-processing of the recorded ENG signal consisted in a digital filtering, spike detection and multi-unit firing calculation. **A**, **B**, **C** are created in BioRender. Valle, G. (2026) https://BioRender.com/lxrugya.

phase of the movement, while only 23.8% (5/21 electrodes) of the recording sites seems to be associated with the extension.

In addition, the firing rates distribution varies across the two directions of the phantom movement. The mean firing rate appears comparable across flexion, 9.58 Hz [interquartile range (IQR), 6.3] and extension, 10.88 Hz [IQR, 12.06] for S1 (Fig. 2D). In S2, the mean firing rates computed across only active channels appear much higher and variable for flexion (median equal to 76.35 Hz and IQR = 111.18), while comparable for extension (median = 9.02 Hz and IQR = 22.03). No significant difference emerges in modulation patterns associated with flexion and extension, concerning the same anatomical region (Friedman test, $n = 56$, flexion-extension: $p = 0.48$) for S1 (Fig. 2A–D).

Instead, for S2, we observed statistical difference from the firing rates peaks during flexion and extension (Friedman test, $n = 56$, flexion-extension: $p = 0.00002$).

In S1, the phantom motion of knee and ankle evoked significant neuromodulation respectively in 72.5% (37/51 electrodes) and 80.4% (41/51 electrodes) of the active recording sites, independently from the direction of the motion (Fig. 2C). Only 23.5% (12/51 electrodes) revealed neural activation concurrently with the phantom movements of the toes, none of which resulted selectively related to the foot, but always coupled with the ankle joint (7 channels), the knee (1 channel) or both (4 channels). We expected knee motion to drive a higher amount of modulation, being more proximal than the other joints, but

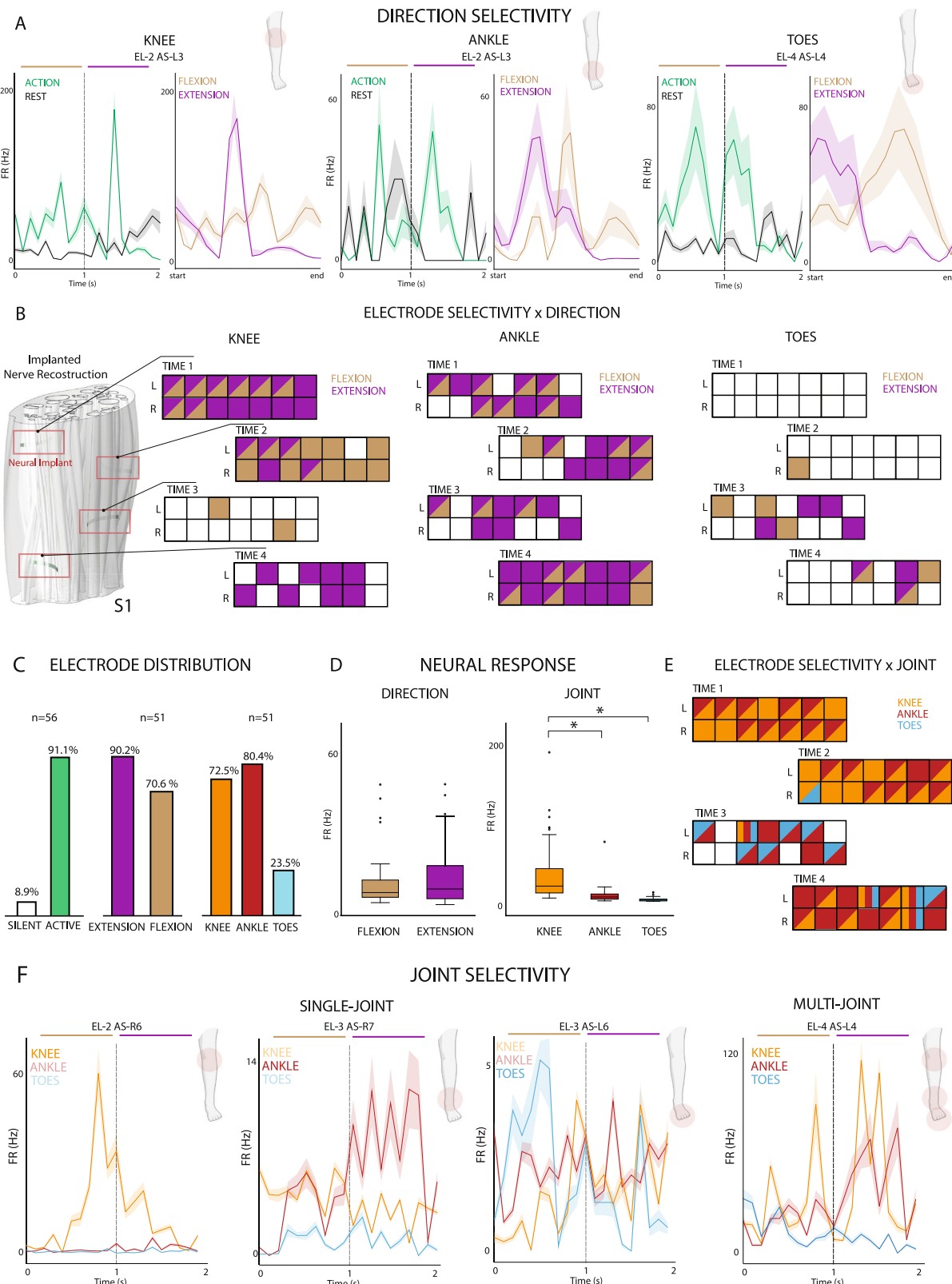

we have to consider how the majority of muscles responsible for knee motion are located above the thigh level and are innervated by the sciatic nerve before the branching of the tibial nerve. In S2, the implanted interfaces appear more selective towards nerve fibers related to the ankle phantom motion, with the number of channels reporting significant modulation for the ankle (85.7%, 18/21 electrodes) almost doubling the ones activating concurrently to toes volitional

movements (42.8%, 9/21 electrodes) (Supplementary Fig. 2A). However, from the reported results we can state that the activity was distributed among all the electrodes, showing both single-joint and multi-joint selectivity.

Furthermore, the firing rates associated with the volitional phantom movements of the different joints reveal distinct activation patterns, with peaks at different frequencies (Fig. 2F). The median firing

**Fig. 2 | Intraneural recording during phantom movements. A** Multi-unit activity of three active sites showing significant direction modulation during flexion (brown) or extension (purple) of the knee, ankle or toes. Mean and SEM are reported. The letters above each plot indicate the TIME number (EL) followed by the active site on the right or left side (L, R). Start and end represent the limits of the 1-s extension phase and 1-s flexion phase present in the same plot. Icons created in BioRender. Valle, G. (2026) https://BioRender.com/lxrugya. **B** 3-D reconstruction of the nerve trunk with its fascicles of participant S1 with the 4 TIME implanted (left). Each of the 4 TIME are reported for the 3 joint movements, showing the significant modulation for each individual channel. L and R indicate the left and right sides of each TIME. A channel colored in purple shows only significant modulation for extension, in brown only for flexion, both colors both directions and white no modulation. Friedman test $p < 0.05$. $n = 56$. **C** Percentage of significant modulation among all 56 channels. Data for silent vs. active, flexion vs. extension and knee vs. ankle vs. toes are reported. $n$ indicates the sample size. **D** Distribution of the average firing rates (FR) computed on all repetitions for each joint (right) and movement direction (left), across the recording sites, revealing neural modulation. On each box, the center line represents the median, the bottom and top edges of the box represent the 25th and 75th percentiles, respectively. The whiskers extend to the minimum and maximum values within $1.5 \times IQR$ from the quartiles. Points beyond the whiskers are shown as outliers. Friedman test *$p < 0.05$ (flex-ext: $p = 0.48$. Ankle-knee, knee-toes and ankle-toes: $p < 0.001$). $n = 56$. **E** Joint selectivity is reported, showing in orange only significant modulation for the knee movements, in red only for the ankle and light blue for the toes. Mixed colors indicate mixed joint selectivity. A white channel indicates no modulation. **F** Multi-unit activity on three different, active sites showing significant single-joint modulation only for a single motion type (single-joint). A channel showing multi-joint selectivity is also displayed (right). Different color lines represent the average signal across all repetition for a specific movement: knee (orange), ankle (red) or toes (light blue) movements. Mean and SEM are reported. Data reported are from S1. Icons created in BioRender. Valle, G. (2026) https://BioRender.com/lxrugya.

rates, computed over electrode contacts, were 24.81 Hz [IQR, 30.62] for the knee, 11.07 Hz [IQR, 6.47] for the ankle and 7.19 Hz [IQR, 2.34] for the toes movements, in S1 (Fig. 2D). Using the nonparametric Friedman test ($p < 0.05$), all the firing rates distributions during ankle, knee and toes movements are pairwise statistically different (Friedman statistical test, $n = 56$, ankle-knee: $p = 0.017$, knee-toes: $p < 0.001$ and ankle-toes: $p < 0.001$). The range of peaks in firing rate seem to be correlated with the insertion level of the muscles targeted by the motor commands of both ankle and toes. In fact, these muscles are active for both types of phantom motion executions but are inserted distally on the foot and proximally on the calf, leading to higher firing rates for the ankle and lower for the toes. This is consistent with the number of joint-selective channels for S1, and with the average response computed on some example channels (Fig. 2A–F).

In S2 (Supplementary Fig. 2B), instead the average peaks patterns computed on active channels, for both joints, are not significantly different (Friedman test, $n = 56$, ankle-toes: $p = 0.275$). The distribution of neural response across joints is characterized by median firing rates equal to 59.9 Hz [IQR, 86.75] and 151.0 Hz [IQR, 68.64], with higher peaks associated with toes and overall higher variability with respect to S1.

In addition, for what concerns the spatial distribution across all electrodes (Fig. 2B–E), in S1 we noticed a separation between sites associated with flexion phase and extension phase of the phantom motion, in particular in knee and toes, where respectively the 32% ($n = 37$) and 17% ($n = 12$) of the modulating sites reveal significant response during both motion directions, while in ankle the overlapping in neural response for the two different phases of the phantom motion interests a wider area of the electrode arrays (46% of the modulating sites $n = 41$, with only 2 electrodes selectively associated with flexion). For knee movements, the number of channels modulating during flexion overall is lower than the number of channels modulating for extension, with also a higher number of electrodes selectively associated with extension (15 channels) with respect to the ones selective to flexion (10 channels). Similarly, for the ankle, a stronger activation of the recording sites during extension can be observed, with only 2 channels selective to flexion and 20 selective for extension; while for toes volitional phantom movements, the neural response of electrodes is balanced between the two directions. In S2, the overlapping between modulating sites appears null for the toes, and minimal for the ankle (Supplementary Fig. 2C). In this last case, only 2 electrodes show significant neural modulation for both directions of the volitional phantom movement, with an evident segregation between channels active for distinct anatomical parts.

## The neural modulation of a specific movement overlaps with its muscular representation

Analyzing the different types of modulation recorded by the 4 electrodes arrays in S1, we can notice how for knee and ankle most channels show significant activation during at least one direction of the motion, with only 33.9% and 26.8% of the total 56 channels being silent during both movements (Fig. 3A). The opposite, instead, can be observed concurrently with toes movements, where the 78.6% of the recording sites tends to not show any significant modulation during both flexion and extension. While for toes movements, the number of sites selectively activated only during flexion appears balanced with the number of sites selectively activated only during extension (both 8.9% of the 56 electrodes), the number of sites with significant modulation in both directions is minimal (3.6%). On the other hand, the number of channels selectively activated during flexion were 17.9% and 3.6% (of the total 56 channels) for knee and ankle movements, respectively, while those only selective to extension 26.8% and 35.7%. We observed a higher number of channels active during both directions in the phantom movement of the ankle (33.9% of the total 56 channels) compared to the knee (21.4% of the total 56 channels). In S2 (Supplementary Fig. 2), both for ankle and toes the number of electrodes not displaying any neuromodulation is significantly higher than the amount of active channels (respectively 67.9% and 83.9% of the total 56 channels are silent, for ankle and toes), which also show a prevalence of recording sites selectively associated with flexion direction of the phantom motion (respectively 25% and 14.3% of the total 56 channels, for flexion in ankle and toes, while only 3.6% and 1.8% for extension in ankle and toes). Also, in contrast with S1, in S2 the overlapping in the electrodes evoking neural activity for the two directions of phantom movement is reduced to only two channels for the ankle and null for toes.

To better interpret these modulations, we considered the overlapping with the muscles involved in the different volitional movements of the lower limb. We only included muscles innervated by fibers belonging to the distal branch of the sciatic nerve[36,37], where the implants were located, as listed in Table S2. After extracting Venn's diagram of the muscular activation and its overlap during phantom movements of the ankle, knee and toes, both for flexion and extension, we transformed them into numerical matrices. We then computed the Herdin distance[38] between these matrices, representing muscular involvement, and the neural activation, as a metric of similarity (0 = highest similarity, 1 = lowest similarity). This metric was exploited in order to provide, beside the qualitative consideration expressed in the previous section, a numerical quantification of the fact that muscular innervation can indeed predict very well the quality in the decodability of the various movements. The results indicated a high correlation between overlapping in muscles participation and neural response during flexion, with Herdin distance equal to 0.15, but a lower similarity between muscular and neural activation during extension, with an Herdin distance of 0.37 (Fig. 3B). Instead, surprisingly the correlation between the neural response during extension phase of phantom motion and overlapping in the flexors' muscles (located in

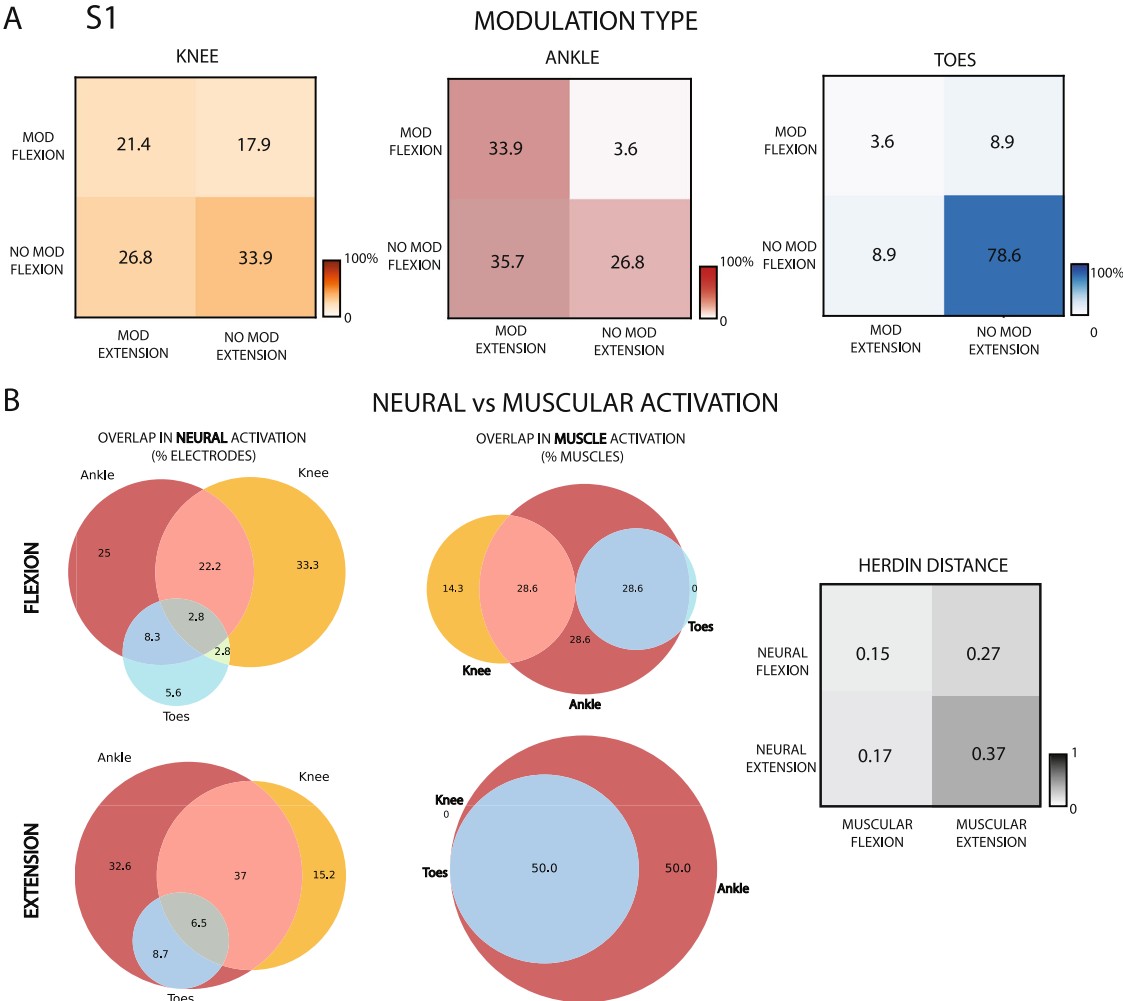

**Fig. 3 | Neural and muscular activations during phantom movements.**
**A** Modulation matrices showing the percentage of electrodes significantly modulating only during flexion, only during extension or both for the three joints. $n = 56$. **B** Percentage of electrodes showing a significant modulation in the neural activation for each direction and joint movement (first column left). $n = 56$.

Percentage of muscles, innervated by the tibial nerve, involved in each movement (second column left). $n = 7$ for flexion, $n = 4$ for extension. Herdin distance between the neural activations sand muscular representations for each movement (right). 0 = highest, 1 = lowest similarity. Data reported are from S1.

the posterior part of the leg) shows high similarity (0.17). This paves the way to the hypothesis that the recorded neural modulation observed during extension could be partially associated with inhibitory response flowing through the tibial nerve and targeting the flexors muscles[39], rather than activation impulses directed to their antagonist extensors muscles (located in the anterior lower part of the leg). However, also spatial overlapping in the electrodes neural responses during flexions shows a medium similarity to the overlapping in the extensors' muscles participation to ankle extension, knee extension and toes extension movements, with an Herdin distance of 0.27. This last result still opens the possibility to have recorded the activity of motor neurons directly innervating extensors muscles.

By cross-referencing the overlapping muscle activations innervated by the distal branch of the sciatic nerve (as detailed in Table S2 and Fig. 3B), we identified two distinct muscle clusters corresponding to different motor synergies: one involving the ankle-toe muscles (flexor and extensor hallucis longus, flexor and extensor digitorum longus) and another involving the knee-ankle muscles (gastrocnemius and plantaris). These clusters align with the electrode joint-selectivity patterns shown in Fig. 2E. Specifically, TIME-1, TIME-2, and the left half of TIME-4 are associated with knee and ankle movements, while TIME-3 and the right half of TIME-4 are linked to ankle-toe control. This spatial segregation of electrode selectivity may reflect a physical separation of

motor neuron fibers targeting different muscle groups at more distal levels of the nerve. Additionally, muscles, such as the soleus and tibialis posterior (ankle) and the popliteus (knee), which also contribute to joint-specific movements and are innervated by the tibial nerve, show selective modulation during phantom movements, further supporting joint-specific functional organization.

### Decoding phantom movements from intraneural recordings using SNN-based decoders

After characterizing the neural efferent response during volitional phantom leg movement conditions, we designed and implemented a SNN to decode phantom movement execution from intraneural ENG recordings. The decoder was trained to classify six classes (ankle flexion/extension, knee flexion/extension, toes flexion/extension) for S1, and four classes (ankle and toes flexion/extension) for S2. The designed decoding architecture used a shallow fully connected SNN composed of fully connected leaky integrate-and-fire neurons (LIF) with synaptic conductance. The input layer comprised 56 neurons, and the output layer had several neurons equal to the number of movement classes. Synaptic transmission was modeled via exponentially decaying conductance. We evaluated this architecture using three different signal encoding strategies. The choice of a single-layer architecture was motivated by the observation that the decoding

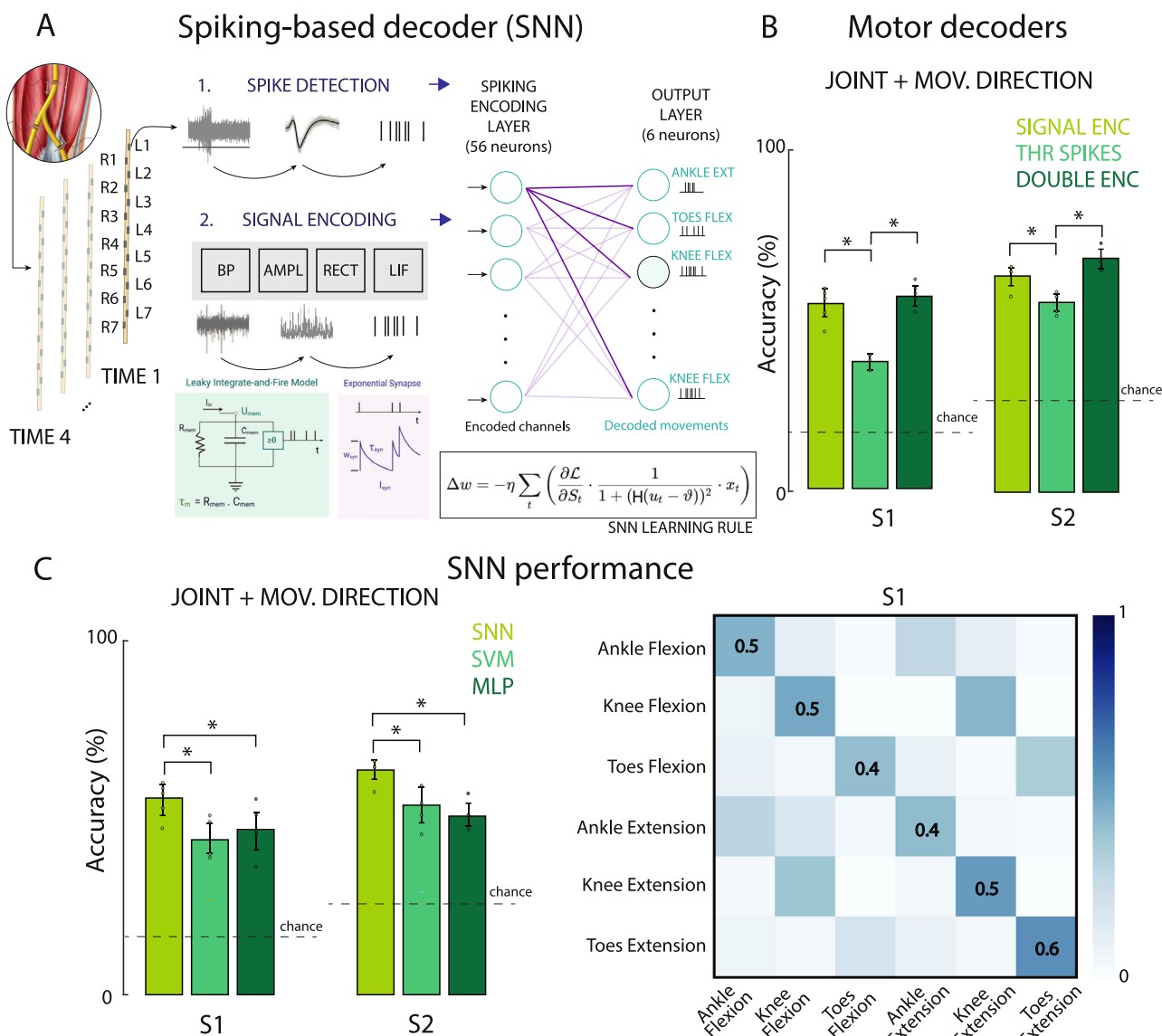

**Fig. 4 | Spiking neural networks to decode phantom movements from the intraneural recordings. A** L is the loss (e.g., cross-entropy between membrane potentials and target class), η is the learning rate, $u_t$ is the membrane potential at time t, ϑ is the spike threshold, and H is the Heaviside function. **B** Comparison of the SNN-based decoders' accuracy in both participants. Performance using the signal standard encoding, threshold spikes and double encoding are reported. Two-sided Paired-t-tests *$p < 0.05$ (Signal enc-thr spikes, $p = 0.003$ for S1, $p = 0.03$ for S2. Double enc-thr spikes, $p = 0.0012$ for S1, $p = 0.001$ for S2. Signal enc- double enc, $p = 0.18$ for S1, $p = 0.06$ for S2. $n = 540$ for S1 and $n = 720$ for S2 (5-fold cross

validation). **C** Comparison of the decoders' accuracy in both participants. SNN spiking neural network, SVM support vector machine, MLP multilayer perceptron. Error bars indicate mean and STD. Chance levels indicated with dashed lines. Two-sided paired-t-tests *$p < 0.05$ ($p = 0.04$ for SNN vs. MLP, $p = 0.0029$ for SNN vs. SVM for S1; $p = 0.006$ for SNN vs. MLP, $p = 0.018$ for SNN vs. SVM for S2). SVM vs. MLP $p = 0.44$ for S1, $p = 0.55$ for S2). $n = 540$ for S1 and $n = 720$ for S2 (5-fold cross validation). Confusion matrix of the SNN performance, using the LIF encoding in S1 for all movement types.

problem was approximately linearly separable, and deeper architectures did not significantly improve performance, therefore, we favored a compact, low-power design aligned with neuromorphic hardware constraints (Fig. 4A). All the reported results are indicated as mean test accuracy ± standard deviation (SD) on 5-fold cross-validation. All the experiments were run training the network using surrogate gradient descend, with the learning rule indicated as in Fig. 4A, on 5-fold cross-validation.

First, we evaluated the decoding performance using an encoder that extracted multi-unit activity from the signals through a thresholding method. This first method returned poor results (Fig. 4B), with 38.86 ± 1.67% of test accuracy on 6 classes for S1 and 55.43 ± 2.77% test accuracy on 4 classes for S2. Therefore, we tested a more innovative spike encoding method[40,41], which was based on the elicitation of the

membrane sensibility of a very simple LIF neuron with no synaptic conductance, which parameters were fine-tuned offline and outside the model. In this way, we aimed to investigate whether motor information was encoded also in fluctuations of the spike rate or actions potentials, rather than being linked to variation in signal amplitude. The events emitted as output by the encoding LIF layer, after injecting as input current a scaled and rectified version of the filtered ENG signal, extracted features related to the power content of the physiological neural activity and its temporal variation. Using this method, we managed to achieve significant higher performance (paired t-test, $p = 0.003$ for S1, $p = 0.03$ for S2), with a test accuracy of 55.14 ± 4.66% on 6 classes for S1 and 63.46 ± 3.21% test accuracy on 4 classes for S2. Finally, we also designed a hybrid double encoding, concatenating the two above-mentioned encoding in a double dimensionality

representation for each data sample. In order to be classified, this last encoding required an adjustment of the previous architecture, going from 56 to 112 neurons in the input layer. The achieved results were statistically higher than the ones obtained using the threshold encoding method (paired t-test, $p = 0.0012$ for S1, $p = 0.001$ for S2), with test accuracies of $58.29 \pm 3.31\%$ for S1, on 6 classes, and $68.29 \pm 3.88\%$ for S2, on 4 classes. Instead, its performance was not significantly higher (paired t-test, $p = 0.18$ for S1, $p = 0.06$ for S2) with respect to using the LIF encoding method, that instead required a network with one-half connections weights to be learned, being therefore more memory efficient.

We compared the results achieved by the SNN decoder on the LIF encoded neural data with two types of more conventional machine learning tools, like a linear support vector machine (SVM) and multi-layer perceptron (MLP), with a hidden layer of 236 non-spiking artificial neurons (Fig. 4C). These methods are commonly used as baseline classifiers in many computational neuroscience studies as motor decoders[42]. Both SVM and MLP were firstly tested using variety of signal features (root mean square (RMS) and power), including also spiking-based features, (such as spike rate), (Supplementary Fig. 3) to find the best possible. For both classifiers, RMS and power were the features achieving higher decoding performance (not statistically significant with respect to each other. Paired t-test, $p = 0.39$ for MLP, $p = 0.79$ for SVM for S1; $p = 0.49$ for MLP, $p = 1.0$ for SVM for S2), while spike rate performed quite poorly in both S1 and S2 (Paired t-test, $p = 0.0004$ for MLP, $p = 0.03$ for SVM for S1; $p = 0.0001$ for MLP, $p = 0.0003$ for SVM). Since the LIF encoding, as explained above, tends to extract features related to frequency power from the signal, we decided to compare the results achieved by the SNN with the performance of SVM and MLP using power as feature (test accuracy equal to $43.70 \pm 5.19\%$ on 6 classes for S1 and $53.75 \pm 5.43\%$ test accuracy on 4 classes for S2 for SVM; while test accuracy equal to $46.85 \pm 6.48\%$ on 6 classes for S1 and $50.83 \pm 3.41\%$ test accuracy on 4 classes for S2 for MLP. No significant difference between SVM and MLP accuracies: paired t-test, $p = 0.44$ for S1, $p = 0.55$ for S2). The SNN achieved significantly higher performance with respect to both SVM and MLP (Fig. 4C; paired t-test, $p = 0.04$ for SNN vs. MLP, while $p = 0.0029$ for SNN vs. SVM for S1; $p = 0.006$ for SNN vs. MLP, while $p = 0.018$ for SNN vs. SVM for S2). Notably, a sparser data representation and computation, which also mimics the natural physiology of information transmission inside biological nerves, results in a better processing of data and overall better efficiency in terms both of decoding performance and memory/energy usage.

Finally, by analyzing the confusion matrix returned on the whole test set by the SNN trained on LIF encoded spike arrays (Fig. 4B) for S1,

we can appreciate how the decoder seems able to correctly identify the 6 classes with accuracies that range from 41 to 57%. Still the network sometimes mistakes the flexion phase of a specific phantom motor task with the extension direction of the same (38%, 44%, 34% of the time, respectively for ankle, knee and toes). Or vice versa, it misleads extension execution of the motion with its flexion counterpart, with the same frequency for the ankle (30% of the time) and slightly less frequently for the knee and toes (39% and 19% of the time respectively). This could be due to neural activation overlapping or variable timing in the execution of the volitional imaginary movement (Fig. 3B). Instead, the errors between different joint-related movements are limited (between 0 and 11%), with the only exception of the extension of the ankle, which gets mistaken for the flexion of the knee 17% of the time.

In S1, overall better accuracies reached for extension are in line with the spatial activation and selectivity of the neural response of the electrodes (Fig. 2B). In fact, having more recording sites selective to extension movements than to flexion-only leads to a 10% difference in test accuracy when distinguishing between the two phases of the same motor task. Surprisingly, despite the low level of recording selectivity in toes phantom movements, toes extension results the class for which the network reaches the highest performance, setting an unseen achievement in the neural decoding of the lower-leg movement. We also considered how the spatial selectivity and movement discriminability might depend on how many TIMEs are implanted into the targeted nerve. To this aim, we calculated decoder performance using individual or multiple TIMEs (Supplementary Fig. 6). In both S1 and S2, we can appreciate how 2 of the 4 TIMEs appear to be more selective than the others (t-paired test, $p < 0.05$) (Supplementary Fig. 6B, top). Also, it's evident how increasing the number of electrodes greatly improves the overall decodability of the attempted actions ($R = 0.93$ for S1; $R = 0.86$ for S2), going from an average classification accuracy of 34% in S1 and 44% in S2 using a single electrode, up to 56% in S1 and 68% in S2 using all four electrodes (Supplementary Fig. 6B, bottom).

## Informing neural decoders with muscular-related signals improves their performance

While processing the ENG data, we filtered the neural signal in a bandwidth between 250 and 7500 Hz as already reported in previous studies[33]. In addition, since the electrodes were implanted in the nerve and located in between the thigh muscles, we also decided to extract inter-muscular (imEMG) information from the recorded activity by filtering the signal between 50 and 350 Hz, frequencies normally related to EMG activity[43].

In S1, we observed a significant modulation in the neural activity, in particular associated to the ankle and knee phantom motion

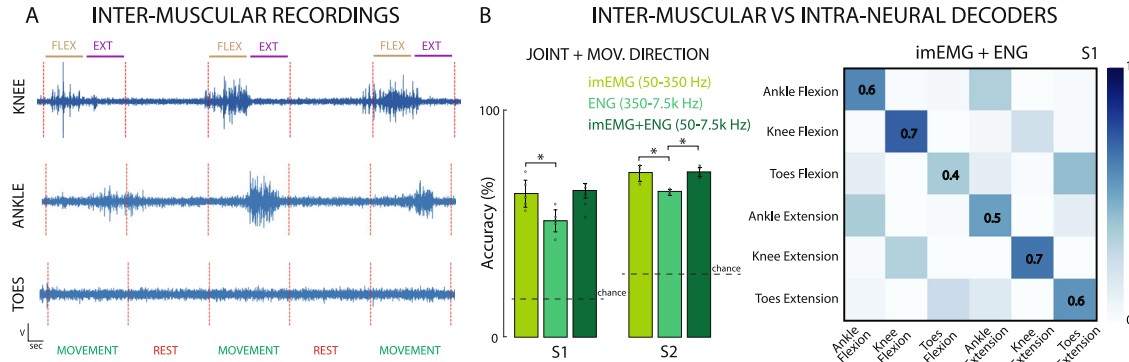

**Fig. 5 | Informing intraneural decoders with inter-muscular EMG signals. A** Example of inter-muscular signals filtered between 50 and 350 Hz. **B** Comparison of the SNN-based decoders' accuracy in both participants using imEMG, ENG or imEMG + ENG bandwidths. Error bars indicate mean and STD. Chance levels indicated with dashed lines. Two-sided paired t-test, *$p < 0.05$ (ENG vs. imEMG, $p = 0.024$ for S1, $p = 0.023$ for S2. ENG + imEMG vs. ENG, $p = 0.06$ for S1, $p = 0.003$ for S2. ENG + imEMG vs. imEMG, $p = 0.77$ for S1, $p = 0.89$ for S2). $n = 540$ for S1 and $n = 720$ for S2 (5-fold cross validation). Confusion matrix of the SNN signal encoder performance in S1 using the imEMG + ENG bandwidth (50–7.5 kHz).

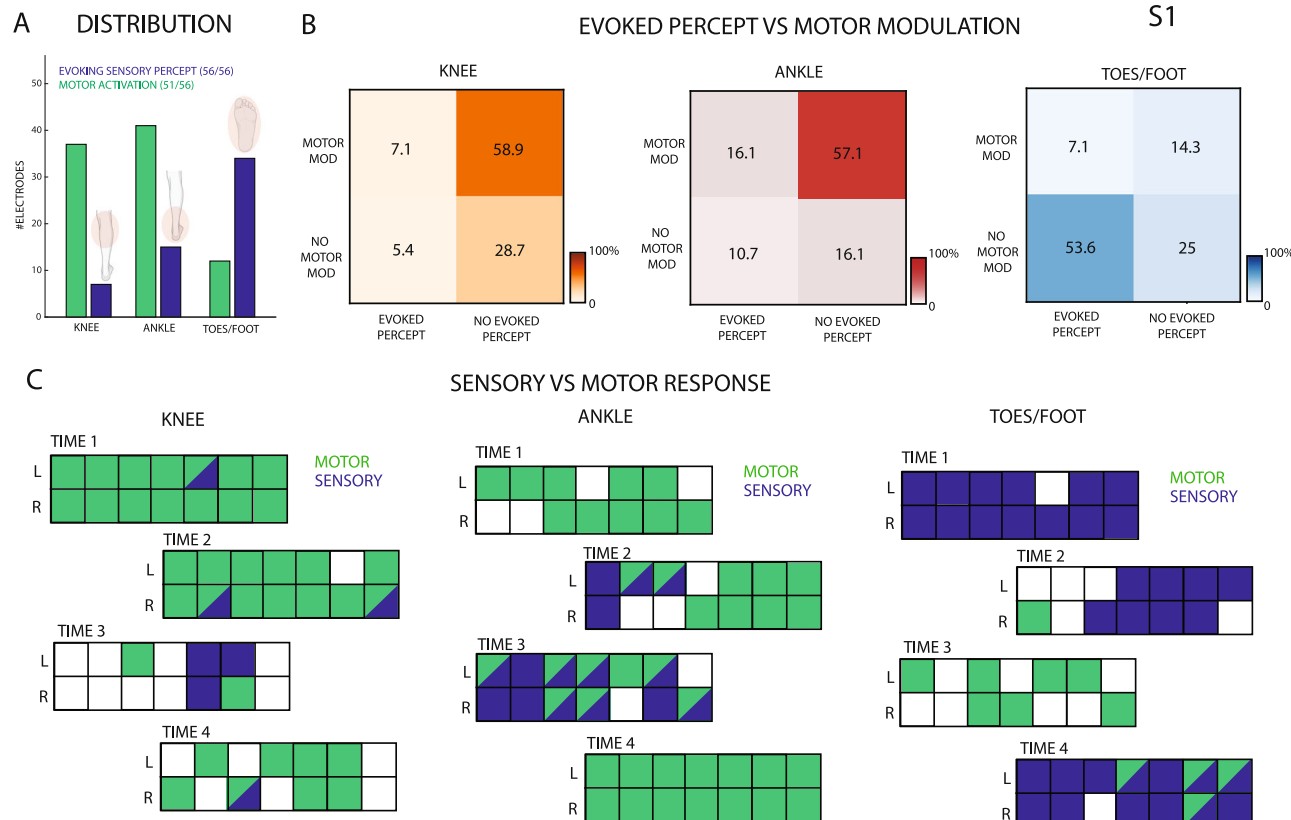

**Fig. 6 | Motor modulation and sensory restoration. A** Distribution of the electrodes showing significant motor modulation compared to those evoking sensation in the knee-calf, ankle-heel or foot-toes areas. Icons created in BioRender. Valle, G. (2026) https://BioRender.com/lxrugya. **B** Modulation matrices showing the percentage of electrodes significantly modulating only during movement, evoking only sensations or both for the three joints. *n* = 56. **C** Each of the 4 TIME are reported for the 3 joints, showing the significant motor modulation (blue) or sensory evoked response (green) for each individual channel. A channel colored in blue shows only significant motor modulation, in green only for evoked-sensations, both colors both responses and white no response. Data from S1.

(Fig. 5A). Giving this imEMG signal as input to our SNN decoder, a performance of 63.14 ± 6.15% on 6 classes for S1 and 72.29 ± 3.68% test accuracy on 4 classes for S2 was obtained (Fig. 5B). Compared to the information contained solely within the ENG-related frequency band (350–7500 Hz), which yielded a test accuracy of 51.14 ± 5.06% across 6 classes for S1 and 64 ± 1.07% across 4 classes for S2, after filtering out all other frequencies potentially associated with residual muscular activity, the decoder trained on imEMG frequency range achieved significantly higher performance (t-test, *p* = 0.024 for S1, *p* = 0.023 for S2). For this reason, we decided to inform our decoders with also muscular signals, by considering the whole frequency range going from 50 to 7500 Hz (ENG + imEMG). The ENG + imEMG resulted in a higher decoding performance than when using ENG frequency band only (Fig. 5B). The increases were +13.2% for S1 and +8.6% for S2 (test accuracy of 64.29 ± 3.26% on 6 classes for S1 and 72.57 ± 2.10% test accuracy on 4 classes for S2; t-test, the increment is not statistically significant for S1: *p* = 0.06, while for S2: *p* = 0.003). The performance was not significantly higher when only the imEMG band was chosen (t-test, *p* = 0.77 for S1, *p* = 0.89 for S2). With respect to the results achieved using only the ENG bandwidth (350–7500 Hz), including the lower frequencies associated with residual muscular activity results in an improved ability to correctly identify the phantom movements of ankle and knee (Fig. 5C). Specifically, the network leverages the imEMG signals from muscles involved in ankle and knee movements to better capture the motion patterns of these anatomical regions and more accurately decode the intended direction of movement. The drawback is a slight decrease in the ability to correctly identify the toes movements, now characterized by much lower amplitudes with respect to the other classes.

We proved the possibility to use nerve interfaces to record not only neural activity related to phantom movements, but also imEMG activation of the residual and above-amputation level muscles, which can be exploited to improve the performance of the motor decoder, also in the absence of the limb.

### Intraneural electrodes allows to record efferent activity and to stimulate afferent fibers

In previous studies on the same patients and using the same intraneural implants[25,26], we showed that it was possible to elicit physiologically-plausible sensory feedback after delivering electrical current with variable intensity, duration and frequency via the implanted TIMEs.

In S1, by delivering current from all the 56 TIME sites independently, it was always possible to evoke artificial percepts (Fig. 6A, B). However, only 12.5% of the electrodes (7 electrodes) targeted sensations related to the calf and knee, and 26.7% (15 channels) perceptions associated with the posterior ankle and heel. 60.7% (34 electrodes) of the total recording sites, instead, evokes tactile sensations associated with the toes and the upper part of the sole. For all three anatomical parts, the overlap between evoked-sensation and significant neural modulation sites is minimal (7.1% for knee, 16.1% for ankle and 7.1% for toes/foot of the 56 total recording sites) and, overall, lower than the number of channels not targeting any efferent nor afferent fibers (28.6% for knee, 16.3% for ankle and 25% for toes/foot of the total 56 channels).

However, while a large proportion of electrodes were responsive solely to volitional phantom movement of the knee/calf (58.9%) and ankle/heel (57.1%) regions, compared to a much smaller fraction

located on afferent fibers targeting the same areas (5.4% for the knee and 10.7% for the ankle), the opposite trend was observed for the toes/foot, where afferent response was more prevalent (Fig. 6A). In fact, 53.6% of the total 56 sites evokes sensations on the sole and toes, while only 14.3% appears able to record activation during attempted flexion and extension. The low motor selectivity of the recording sites is also associated with a clear spatial separation between electrodes recording efferent motor neurons with respect to channels implanted close to afferents somatosensory fibers (Fig. 6C). This is particularly evident for the foot, but also true for ankle/heel and knee/calf where the overlapping between motor selective electrodes and AS eliciting sensations is limited. This suggests that the afferent and efferent fibers in the tibial nerve result to be already segregated within the nerve at the implantation level, before the bifurcation of the sciatic nerve into tibial and common peroneal (Fig. 1B).

The spatial separation between sites selective to afferent and efferent fibers is confirmed also in S2 (Supplementary Fig. 4B, C), with an overlapping of only 12.5% and 5.3% of the total 56 recording sites, respectively for ankle and toes. The distribution of the electrodes eliciting percepts appears still mirroring the one seen in S1, with the number of sites associated to sensations of the foot/sole higher than the ones related to the ankle/heel (Supplementary Fig. 4A). In this case, not all the electrodes, if independently stimulated, were able to evoke a percept (35/56). Additionally, 60.7% of the 56 channels for the ankle/heel region and 48.2% for the toes/foot region are not involved in either evoking sensations or recording motor modulation (Supplementary Fig. 4C).

## Discussion

### Intraneural interfacing allows the access to intrinsic muscles signals

In this study, we demonstrated the feasibility to record movement-related information from signals recorded from inside the leg nerves. The information, both flexion/extension-related and joint-related, was captured by the implanted electrodes. Given the ability of the amputees to generate motor activity associated with attempts to control a missing limb movement within a short period of time, we infer that the motor central and peripheral pathways remain largely intact from the functional point of view, as shown by imaging studies[44]. This is in line with what observed on the sensory pathways, given the easily identified, discrete and graded sensations elicited by direct electrical stimulation of the nerve[25], without the need for any sensory training.

The use of 4 TIMEs in a single nerve with a diameter of around 1 cm allowed the recording of significant modulation for each of the performed movements in both participants (S1: knee-ankle-toes, S2: ankle-toes). This result indicates an appropriate coverage of the cross-sectional nerve space with this number of penetrating multi-channel electrodes. Indeed, not all the joints' movements were recorded on every single electrode (e.g., in S1 TIME-1 and TIME-2 showed modulation during knee and ankle movements, while toes-related movements were detected mostly on TIME-3 and TIME-4), supporting the appropriate selectivity with 4 implants. This electrode placement has also been suggested for stimulation purposed using realistic computer simulations[45,46], considering the nerve fascicular topography[47].

Intriguingly, in the recorded signals, we observed neural activity that could still be associated with extension movements. We propose two non-mutually exclusive explanations for this finding. First, the recording site was located above the bifurcation of the sciatic nerve, before the separation into the tibial and peroneal branches. At this level, some fibers from both branches are likely intermingled, which could explain why extension-related signals were still observed within a predominantly tibial recording site. Second, the altered neural control mechanisms following amputation may contribute to overlapping activation patterns. In intact physiology, reciprocal inhibition and common drive mechanisms coordinate agonist–antagonist muscle

pairs[48–50]. However, in amputees, the absence of proprioceptive feedback may disrupt these inhibitory loops, leading to concurrent activation of both flexor and extensor pathways during attempted extension. This interpretation is consistent with previous evidence of nonspecific or co-contracted phantom movements[51,52]. Together, these findings suggest that both anatomical factors and altered spinal circuitry after amputation may underlie the reduced separability of neural patterns during the execution of flexion-extension phases of the same movement.

The type of modulation response observed in the implanted nerves was mostly an increase of the multi-unit nerve activity (firing rates) locked to the phantom movement onset or offset. The average aggregate firing rate was comparable to that reported in studies of upper-limb nerve intraneural recording[29,33] in humans during hand movements. However, in the case of more distal joint-related movements, we observed a decrease in nerve firing, possibly due to the proximal electrode location (thigh level) and to the different number and size of the muscles, generated force, and number of recorded efferents, involved in that isotonic movement[53,54].

In addition, thanks to the intimate contact of this penetrating neurotechnology with the nervous fibers, it was possible to capture the neural modulation in response to different movements of the phantom limb in both amputated individuals. Specifically, a significant modulation was observed, not only for proximal limb movements, such as knee flexion and extension, but also to foot-related movements. Notably, movements of the toes are controlled by intrinsic leg and foot muscles, such as the flexor hallucis longus (for flexion of the great toe) and flexor digitorum longus (for flexion of toes 2-5), that are lost after a so proximal limb amputation. This is also the case for muscles controlling the ankle flexion (also known as dorsiflexion) and extension (plantarflexion), such as the plantaris and soleus (Table S2). This reported evidence shows the potential of such an approach that allows to access to information, not available to noninvasive[55] or other surgical approaches targeting only residual muscle signals[16].

### Spiking-based decoders outperforms conventional decoders for neural signals

In this study, we implemented and tested SNN-based decoders able to extract phantom movement execution from intraneural ENG recordings. The SNN significantly outperformed conventional non-spiking decoders. The performance advantages stem from their ability to leverage the temporal precision and discrete nature of action potentials, which carry rich, high-fidelity information about neural intent. Unlike conventional decoders that typically rely on averaged firing rates or continuous signal features (e.g., RMS or low-frequency envelopes), spiking-based approaches preserve the fine-grained timing of neural events.

This allows them to capture rapid changes in neural activity and subtle variations in motor commands that are essential for precise and responsive control. Furthermore, spiking activity tends to be more robust to noise and drift compared to low-frequency field potentials[56]. These properties make spiking-based decoding particularly well-suited for applications in neuroprosthetic control, where both accuracy and responsiveness are critical. Moreover, our encoding method, also based on leaky-integrate and fire neuron dynamics, enhanced the separability of ENG signals by transforming raw neural activity into event-based spike trains. This encoding leverages the integrative properties of spiking neurons to emphasize temporally stable features while attenuating high-frequency noise, critical for handling the low-signal-to-noise ratio conditions typical of intraneural recordings. The synergy between spike-based encoding and spike-based decoding results in a coherent, noise-resilient pipeline that improves classification robustness.

The decoder was intentionally designed as a shallow, single-layer network of LIF neurons. This choice was driven by two key

observations: i) the decoding task was approximately linearly separable, as confirmed by the strong performance of linear SVMs, and ii) deeper networks, while theoretically more expressive, tend to overfit and generalize poorly in small datasets[57]. The single-layer architecture offered an optimal trade-off between complexity and generalization, achieving high decoding accuracy with minimal training data and parameter overhead. Additionally, the proposed SNN-based decoder is also well-suited for implementation on neuromorphic hardware, which mimics neural architectures and processes spikes natively[35]. This makes it feasible to embed the entire decoding process directly into a prosthetic device, enabling real-time, onboard motor control without the need for external computing systems. Such integration not only improves the usability and autonomy of neuroprosthetic systems but also brings them closer to clinical viability. Overall, the use of a compact, biologically inspired SNN decoder enabled accurate and efficient decoding of volitional lower-limb phantom movement execution from intraneural signals, providing a preliminary validation of neuromorphic approaches for motor decoding. These approaches require future real-time and chronic testing.

## Implantable intraneural electrodes allow for a bidirectional neural interfacing

The here reported intraneural electrodes showed to enable a bidirectional interfacing with peripheral nerves. We showed a proof of concept for using intraneural recording to decode phantom movement executions from the recorded signals, and to evoke informative tactile sensations in lower-limb amputees[26]. The recorded neural modulation was mostly in response to knee and ankle movements, while less channels showed modulation during phantom foot movements. On the other hand, the sensations evoked by directing stimulating the nerves were more often reported on the plantar area of the foot compared to the lower leg. This result respects the muscular and cutaneous innervations of the tibial nerve (Fig. 1B). Indeed, more afferent fibers innervate the plantar region of the foot[58], and thus it will be easier to activate them in the nerve, evoking a sensation on the foot. Considering their placement within the distal branch of the sciatic nerve, we observed limited cluster and somatotopic organization of motor and sensory fibers at the thigh level. This is in line with analyses on the fascicles distribution within somatic nerves, in which the proximal segment corresponds to a region where, as demonstrated by ChAT+ immunostaining, motor axons are heterogeneously distributed, and no prediction of their destination can be established a priori[59].

Thanks to the location of the implants at the thigh level, this neural interfacing allowed the recording of both neural signals and also the EMG-related signal, working as inter-muscular electrodes. Similar EMG spiking has previously been reported for intraneural recordings in individuals with upper-limb amputation[60]. To maximize the decoding performance, we showed that including the band of the imEMG signal could potentially improve the decoding accuracy, allowing for a more stable control of multiple DoF, avoiding additional surgical intervention as TMR, AMI or RPNI[11]. The combined use of ENG and imEMG provided muscular recordings with higher signal quality than surface EMG, potentially reducing crosstalk and impedance variability. This suggests that surface EMG alone may not be sufficient to distinguish among these phantom leg movements, though this requires dedicated validation. Importantly, these implantable interfaces could serve dual roles for both recording and stimulation, enabling stable, bidirectional control without electrode repositioning. While SNN performance was similar for imEMG and imEMG + ENG, ENG may capture complementary information from muscles no longer active post-amputation, potentially improving decoding.

In practical applications, residual thigh muscle activity unrelated to phantom movements, recorded by the implants, could interfere with decoding stability. While in the controlled setting, this effect was limited, in daily use, amputees may recruit the same muscles for other tasks, potentially contaminating the motor decoding. Incorporating ENG-based signals could help disentangle neural from residual muscular activity by filtering out unrelated EMG components. However, future studies in more natural, dynamic scenarios are needed to investigate this effect.

In the context of bionic legs, this direct interfacing using TIMEs holds significant promise for allowing the bidirectional neural connection between the human peripheral nervous system and a prosthetic device. This would potentially support: i) the restoration of both high-resolution tactile feedback and direct neural control; ii) a direct access to intrinsic muscular information and thus to extract all the volitional movements of the users; iii) a single implanted technology. This might improve not only prosthesis functionality and control accuracy, but also device embodiment and the overall user experience in individuals with limb loss. An important next step will be to test whether different speeds and torques of phantom movements can be reliably decoded from intraneural signals, building on preliminary findings in the upper limb[30]. Realistic prosthetic control also requires handling co-contraction and joint stabilization[61]. Addressing these factors will be critical for translating intraneural decoding of phantom movements into functional, real-time prosthetic control.

## Limitation of the study

There are some limitations to this study that should be considered. First, the small sample size limits how broadly the findings can be applied. While the results are encouraging, more participants would be needed to confirm that the approach works consistently across individuals and amputation levels, especially given the variability in nerve anatomy, implantation and neural signal quality.

From these findings, we cannot predict long-term outcomes when decoding phantom movements. Signal stability may deteriorate over extended periods due to factors, such as tissue responses surrounding the electrode, minor positional shifts, or nerve adaptation, as previously reported for TIME implants in humans[62,63]. Importantly, while the use of TIMEs eliminates the need for nerve rerouting procedures, such as TMR, RPNIs, or AMI, their penetrating intraneural nature introduces challenges related to fibrotic encapsulation and chronic tissue response. These biocompatibility aspects critically influence long-term signal reliability and selectivity and warrant careful consideration when comparing TIMEs to implant-independent approaches. Dedicated longitudinal studies are therefore required to fully assess chronic stability and functional durability. Moreover, future studies should also consider implementing spike sorting or duplication removal to more accurately estimate firing rates and account for multi-channel detection of the same units.

Additionally, the electrodes were implanted in only a single peripheral nerve, which constrains the amount of motor information that can be accessed. While this simplifies the procedure and reduces risk, it also limits the potential for more complex or naturalistic movement decoding and ultimately prosthetic control. Implanting both the tibial and the peroneal breaches of the sciatic nerve could potentially expand the repertoire of decodable limb movements.

Finally, although we demonstrated motor decoding, this was done in an offline environment. This is only a proof-of-concept of using intraneural signals to decode phantom movements. Real-time use would introduce more variability and noise, which could challenge the decoder's performance. The robustness of the decoding algorithms under dynamic, real-world conditions remains unknown. Moreover, the use of the same electrodes also for delivering neurostimulation to restore sensory feedback would introduce stimulation artifacts. Techniques based on blanking[64] or time-division multiplexing[65] should be implemented to allow a stable bidirectional configuration. Future work should explore how the neural decoding holds up under more dynamic and bidirectional conditions.

A final consideration concerns the nature and execution of phantom movements in the amputee participants. Both individuals reported vivid phantom limb perception and voluntary control of their phantom legs, as previously described[25]. However, in the absence of proprioceptive and visual feedback, the effective timing and specificity of these movements cannot be objectively verified. As a result, the neural and muscular activation patterns may include nonspecific contractions of the residual limb rather than articulation-specific motor commands[52,66,67]. This lack of sensory feedback may contribute to the variability and reduced separability observed in the decoding of phantom movements, particularly at higher imposed rhythmic rates. These aspects highlight the intrinsic limitations of interpreting motor-related neural signals in the context of deafferented sensorimotor systems.

## Methods

### Participant recruitment and ethics
Two unilateral transfemoral amputees were included in the study. All of them were active users of passive prosthetic devices (Ottobock 3R80) (Table S1). Every experiment involving human participants has been carried out following a protocol approved by an ethical commission. Ethical approval was obtained from the institutional ethics committees of the Clinical Center of Serbia, Belgrade, Serbia, where the surgery was performed (ClinicalTrials.gov identifier NCT03350061). Each participant gave informed written consent, and they did not receive compensation for the study participation. During the entire duration of our study, all experiments were conducted in accordance with relevant EU guidelines and regulations.

### Surgical procedure
The electrode implantation procedures were conducted under general anesthesia. A longitudinal incision was made along the sulcus between the biceps femoris and semitendinosus muscles, positioned at the midpoint of the posterior aspect of the thigh and beginning approximately 4.5 cm proximal to the end of the amputation stump. To expose the sciatic nerve, the semitendinosus muscle was retracted medially while the biceps femoris was moved laterally. Each participant received 4 TIME-4H electrodes. For each electrode, the surgeon created a small window in the epineurium, allowing transverse passage through the visible fascicles. The electrode was then drawn through the nerve so that its active (stimulating) sites made direct contact with the fascicular tissue. Once positioned, the electrode was secured to the epineurium using sutures through the fixation tabs. After all electrodes were implanted, a fascial flap was raised by cutting and folding a patch of fascia around the electrode cables, which was then sutured to the underlying tissue for stabilization. The electrode leads were tunneled subcutaneously through the thigh and exteriorized via small incisions made on the anterolateral aspect of the thigh, a few centimeters below the iliac crest, enabling transcutaneous connection to an external neurostimulator.

During surgery, electrode impedance was continuously monitored using the stimulator's built-in impedance check function to ensure proper contact and functionality. Following implantation, each AS was tested, and only contacts with impedances below 100 kΩ were considered functional, confirming their capacity to deliver current to the nerve. Each surgery lasted approximately 4 h. At the conclusion of the study, all implanted electrodes were surgically explanted from both participants. This study was performed within a larger set of experimental protocols aiming at assessing the impact of the restoration of sensory feedback via neural implants in leg amputees during a 3-month clinical trial[24–28]. The data reported in this manuscript was obtained in multiple days during the 3-month trial in two leg amputees. Detailed characterization of tissue–electrode interactions and biological responses at the nerve level following implantation of TIME electrodes have been reported in the same previous work[62]. In

particular, the histological analysis performed after explantation (90 days) revealed that the intraneural electrodes generated a foreign body response (FBR) (Supplementary Fig. 1G). A significant FBR was only present at the implantation level, with the proximal and distal part of the nerve remaining unaffected given the low number of macrophages. No severe adverse events (e.g., infections) have been observed during the trial, while perceptual thresholds, sensation location changes and some electrode failures have been reported. The exact placement of the TIMEs relative to motor fibers is unfortunately undetermined. However, modeling studies[45,46] suggest it is possible to estimate the optimal number of TIMEs required to cover the cross-section of a target nerve.

### Intraneural electrodes and recordings
Each of the TIMEs (latest generation TIME-4H) implanted in the two amputees was constituted by 14 AS and two ground-electrodes. Details concerning design and fabrication can be found in ref. 68. For each subject, 56 electrode channels were then accessible for recording and stimulation on the distal branch of the sciatic nerve[62]. The neural recorder was the Grapevine neural Interface System (Ripple, LLC), which is a commercial device that can be used for the recording of neurophysiological data through up to 64 high-impedance microelectrodes, divided into 2 ports of 32 poles each. Therefore, up to 4 electrodes (56 AS) can be recorded simultaneously and digitally sampled at 30 kHz, and each port addressed two TIMEs. Each electrode includes 14 capacitively coupled active outputs (channels-AS) and 2 non-capacitively coupled references.

### Experimental protocol and movement tasks
The participants were sited, and they were asked to attempt movements of the knee, ankle and toes with their phantom limbs, as shown on a screen. The type and the timing of the task (start time and end time) were specified using a real-time custom interface designed in MATLAB (R2016a, The MathWorks, Inc., USA) positioned in front of the participants. We asked the participants to perform six different limb movements: knee flexion, knee extension, ankle dorsiflexion, ankle plantarflexion, toes flexion and toes extension (Supplementary Fig. 5). Each movement was randomized in 3 blocks and repeated 10 times. The subjects had to move their phantom limbs for each trial as required; one trial lasted 2 active seconds and was followed by 2 s of rest (no movement), seen as an inter-pause between two successive repetitions of the same phantom movement. The total movement consists of 1 s for the flexion, immediately followed by 1 s for extension. Here we reported recordings in 2 sessions for S1, 13 and 90 days after implantation and 1 session for S2, 90 days after implantation. Data recordings of S2 during the knee movements were found to be corrupted during the registration process and were therefore excluded from the analysis.

### ENG pre-processing analysis
Collected ENG data were first analysed in MATLAB (R2024a, The MathWorks, Inc., USA) and then processed using Python (version 3.11.9, the Python Software Foundation) in the Visual Studio Code (VSCode) environment. We applied the detailed analysis offline, as following:

Corrupted file removal. Firstly, we manually discarded some of the recorded files, since they were associated with flat, cut paths or massive artifacts. In particular, in S2, data related to knee movements were found to be corrupted during the registration process and were therefore excluded from the analysis.

Temporal filtering. Raw ENG data from 56 AS were pre-processed with by applying first an infinite impulse response (IIR) notch filter at the multiples of 50 Hz (between 0 and 10 kHz) with a quality factor of $Q = 100$, to remove any interference due to power line-effect, which are prone to appear at periodical frequencies. We then removed signal

drift and nonneural frequency content by designing and using a 4th order Butterworth band-pass filter between 250 and 7500 Hz, in accordance with preceding studies processing intraneural recordings in upper-limb amputees[33].

Inter-muscular activity. To assess the amount of information related to the imEMG activation in the recorded signal, we substituted the temporal filtering in the previous step with at 4th order Butterworth band-pass filter between the EMG-specific bandwidth 50 and 350 Hz. We called the so filtered neural signal imEMG path. We also compared the classification performances achieved on the imEMG signal with a version of the ENG signal filtered between 350 and 7500 Hz, so by only considering the information related to the neural activation and completely discard the muscular residual content (called ENG in Fig. 5). Finally, we also considered the signal recorded filtered on the total bandwidth (between 50 and 7500 Hz), including both inter-muscular residual activity and modulation associated uniquely with the axons, therefore called imEMG + ENG.

### Characterization of intraneural spiking activity

Spike detection. To characterize the evoked neural activity, as an electrode-specific metric of the modulation during motion as opposed to the inter-actions resting condition, we performed a multi-unit spike detection, involving a simple subject-adaptive, automated thresholding. After a keen tuning, the threshold value was set equal to a factor, respectively to 3.5 and 4 for S1 and S2, multiplied by the SD of the signal[69]. The threshold was determined separately for each recording channel.

Motor maps extraction. From the obtained spike-event times from each electrode, we drew the firing rates by summing the number of spikes appearing in nonoverlapping temporal bins of 100 ms duration (Fig. 1D). As a quantitative metric of the neuromodulation, for each electrode, we computed the z-score, as the difference between the mean firing rates during action ($\mu_{SR_{actions}}$) vs. rest ($\mu_{SR_{rests}}$) divided by the SD of the firing rate during rest ($\sigma_{SR_{rests}}$):

$$Z - \text{score} = \frac{\mu_{SR_{actions}} - \mu_{SR_{rests}}}{\sigma_{SR_{rests}}} \quad (1)$$

If the z-score exceeded the value of 0.5, we assumed neuromodulation associated to a specific phantom motion present in the considered site.

### Spike-based encoding

We tested a method to encode the information contained in the analog signals into spike arrays, to be input into the SNN encoder, based on the elicitation of the membrane sensibility of a simple LIF neuron. This method was implemented on simulation using the Brian2 software[70]. Before providing the pre-processed signal to the neuron, it was full-wave rectified in order to preserve both the positive and negative signal phases and scaled using z-score normalization. The dynamics of the membrane of this neuron can be modeled as:

$$\frac{dU}{dt} = \frac{U[t-1] - U_{\text{drive}}[t]}{\tau} \quad (2)$$

Where $U$ is the LIF neuron membrane potential, $U_{\text{drive}}$ is the continuous domain ENG signal, acquired by a specific channel, and $\tau$ is the time constant of the membrane. If no input is injected into the neuron, the membrane potential tends to decay in an exponential way in time, with time constant $\tau = 10$ ms, till it reaches its resting state value. If the membrane of the neuron reaches a specific threshold value (in our case set to thr = 350mV), the neuron emits a spike/event as output and the membrane potential is reset to a specified value ($U_{\text{reset}} = 0$mV). The simulation was carried out for the entire duration of the recording, and we employed a refractory time of 1 ms, in which the LIF neuron cannot

spike even if the threshold is reached. This type of event-based encoding extracts information related to the power contained in the signal, and its rate of variation in time in a biologically-inspired way that resembles how the human body processes sensory information (like for example in the cochlea[71,72]). This encoding was later binned using 2.5 ms nonoverlapping windows (the same length of the simulation time step), in order to reduce the dimensionality of the generated event-representation.

### Conventional motor decoders

We employed two traditional machine learning classifiers as baseline motor decoders: a SVM with linear kernel and an MLP. At first, we also tried implementing an SVM decoder with radial basis function (RBF) kernel, to assess if the translation of the data in a nonlinear alternative feature representation could improve classification, but we eventually realized that the linear SVM always outperformed the one with RBF kernel, which was then discarded for further analyses. Instead, we designed MLP with 112 neurons in the input layer and 236 (210% of the number of input neurons) in the hidden layer. The output layer dimensionality was set to 6 for S1 and 4 for S2, in compliance with the classification task.

### Feature extraction for conventional decoders

The SVM and MLP classifiers were trained on signal segments of 100 ms as input data samples. For each data sample, we computed the following features, separately for each channel:

RMS:

$$\text{RMS} = \left( \frac{\sum_{i=1,..,N} x_i^2}{N} \right)^{\frac{1}{2}} \quad (3)$$

Power:

$$\text{power} = \frac{1}{N} \sum_{i=1,..,N} \|X\|^2 \quad (4)$$

Spike rate (SR): as the sum of spikes (obtained using the same thresholding method seen for the characterization and quantization of the spike activity) normalized on the window length.

Each of these features were computed on non-overlapping temporal windows of 50 ms. Then, the two values obtained independently for each electrode were concatenated in a single feature-vector representation of the data sample $\in \mathbb{R}^{112}$.

### Shallow spiking neural network (SNN)

We designed a single-layer SNN comprising 56 input neurons (or 112, when processing the signal represented using the double encoding, which combined both the thresholding and the LIF-neuron encodings) densely connected to the output layer. The output layer consists of 6 for S1 or 4 for S2 LIF-neurons, encoding the classification assignment for each data sample. Neurons receive the encoded spike trains from the 56 ENG channels, once removed the grounds. The network is designed and trained using SNNTorch[73] (version 0.9.1), a Python package that extends the capabilities of PyTorch (version 2.3.1), taking advantage of its GPU-accelerated tensor computation and applying it to spiking neurons. Neurons are simulated as second-order LIF neuron model, using the synaptic class already available on SNNTorch, which considers also the synaptic conductance, modeling the synapses between neurons as exponential kernels.

When an input spike arrives, the synaptic current is incremented by the corresponding synaptic weight. Subsequently, the synaptic current decays exponentially over time with a time constant denoted by $\tau_{\text{syn}}$. The neuron integrates the incoming synaptic currents in the membrane potential $U$, which instead leaks over time with a time

constant $\tau_{mem}$. Once $U$ reaches a threshold value $U_{thr}$, that we set equal to 0.5, the neuron emits a spike, $S_{out} = 1$, and $U$ is reset. The equations expressing the behavior of the network can be summarized by:

$$I_{syn}[t] = \alpha I_{syn}[t-1] + I[t] \qquad (5)$$

$$U[t] = \beta U[t-1] + I_{syn}[t] - S_{out}[t-1] U_{thr} \qquad (6)$$

$$S_{out}[t] = 1 \text{ if } U[t] \geq U_{thr} \text{ else } 0 \qquad (7)$$

Where $I_{syn}$ represents the synaptic current, $I$ is the input current, $U$ is the membrane potential of the neuron, $U_{thr}$ the threshold value of the membrane potential and $S_{out}$ a binary value representing the output at the indicated time instant. The last equation indicated the spike generation mechanism, while the term $S_{out}[t-1]U_{thr}$ in the second equation represents the reset mechanism. $\alpha = e^{-\frac{\Delta t}{\tau_{mem}}}$ and $\beta = e^{-\frac{\Delta t}{\tau_{syn}}}$ reflect, respectively, the synaptic current and membrane potential decay rates, whose biological shape is simulated using exponential functions. Both are defined as functions of the respective time constant. We fixed the value of $\alpha$ and $\beta$, respectively, equal to 0.95 and 0.98.

The network is simulated using a time step of 2.5 ms, after binning the original spike arrays as described in the spike encoding section of the methods. For what concerns the training of the network, to address the non-differentiability associated with the Heaviside function, which represents the spike generation mechanism, synaptic weights and neuron threshold are trained using the Heaviside function as surrogate gradient in the backward step[74]. We trained the network using mean square error spike count loss function and Adam optimizer, with learning rate $lr = 10^{-3}$. This loss encourages the target neuron to fire at a specific target frequency, while simultaneously inhibiting the firing rates of all the other output neurons to another value. The target frequency is a critical hyperparameter that requires adjustment based on the selected input time window width. Since we used data samples of 100 ms (downsampled at 750 Hz), we used a target firing ratio of 0.8, which means that we expect the correct output neuron to fire at each time step with a probability of 80%. Likewise, we set the incorrect firing rate for nontarget output neurons to 0.001 to reduce excessive modifications in the weights. As a performance measure we used the accuracy rate function in SNNTorch, which first predicts as the correct label the one associated to the output neuron with the higher firing rate, and then compares it with the target after being converted into a one-hot-encoding. The network was trained on 500 epochs, using a batch size of 70 data samples.

### Single and multi-electrode contribution in SNN decoding

In order to investigate how the amount of information recorded by each TIME was involved in the overall decodability of the attempted movements, we also evaluated a variation of the SNN model, which used a reduced number of input neurons (from 56 to 14), separately on each TIME. We used a 5-fold cross-validation approach, reporting the averaged test accuracies across the different folds (mean ± SD) for both S1 and S2 (Supplementary Fig. 6B). The performance on different TIME electrodes, on the same data split, were compared using t-paired test.

Additionally, in order to assess how the decodability was influenced by the number of electrodes, we performed analogous analysis using increasing number of TIMEs. We trained a SNN with 28 and 32 input neurons, respectively, on all the possible combinations of 2 and 3 TIMEs. For each combination, we computed the average test performance using 5-fold cross validation. Finally, the accuracies were also averaged across all the combinations (Supplementary Fig. 6C), comparing them with the average across single TIMEs as well as the accuracy achieved using all 4 TIMEs. We statistically compared

performances, and we calculated Pearson's correlation coefficient (R) to quantify the effect of adding multiple electrodes.

### Sensory maps

Sensory maps were constructed based on the projected fields maps evoked via intraneural stimulation for each participant. The same 56 electrode channels used for recording were also accessible for stimulation. The electrodes were connected to an external multichannel controllable neurostimulator, the STIMEP (Axonic and University of Montpellier). The scope of this procedure was to determine the location of the electrically-evoked sensation, as described by Petrini et al.[25]. When the participant perceived any electrically evoked sensation at the minimum charge (i.e., perceptual threshold), the location was reported by the participants. This was repeated five times per channel and then averaged. All the data were collected using a custom-designed psychometric platform for neuroprosthetic applications, which allowed us to collect data using a standardized procedure[75]. These maps were used to calculate the number of active sites evoking sensations in the knee/calf versus ankle/heel versus foot/toes areas for both S1 and S2.

### Herdin's correlation matrix distance and muscle groups

The Herdin's correlation matrix distance[38] is the distance between two correlation matrices R1 and R2 as defined by:

$$d_{corr}(R_1, R_2) = 1 - \frac{tr\{R_1, R_2\}}{\| R_1 \|_f \| R_2 \|_f} \in [0, 1] \qquad (8)$$

Where tr{.} denotes the matrix trace and $\|.\|_f$ the Frobenius norm. It becomes zero if the correlation matrices are equal up to scaling and one if they differ to a maximum extent. For the justification and details, see the original paper.

In our analyses involving muscles, we only considered those innervated by the sciatic nerve or its branches below the amputation level, therefore excluding the muscles of the thigh (Table S2). Also, we are excluding the muscles involved in joint movements different from the one considered in our experimental set-up (inversion of the foot, eversion of the foot, etc.), such as the Peroneus Brevis, only involved in the eversion of the foot, and the Peroneus Longus, only weakly involved in the plantarflexion and engaged mainly in the foot eversion, both innervated by the Superficial Peroneal Nerve. Finally, we consider the extrinsic muscles of the foot, responsible for actions, such as plantarflexion, dorsiflexion and general flexion or extension of the toes, not considering the intrinsic muscles, located in the foot and responsible for finer motions of the individual digits. These are innervated by the Plantar Nerve, a distal branch of the tibial nerve.

### Statistical analysis

All data were exported and processed offline in MATLAB (R2024a, the MathWorks, Inc., USA) and reported as mean value ± SD (unless elsewise indicated). All statistics were performed using the available built-in functions. Since all the indicated decoders performance (SVM, MLP, and SNN) are the mean performance on 5-fold cross-validation using the same data split, we compared the achieved test accuracies using pairwise t-test. Additional, statistical tests were applied in order to compare the difference between the distribution of the firing rates (averaged on all the repetitions of a specific phantom movement on each recording site) of the different phantom movements and directions. The distributions represented in the boxplot (Fig. 2D) are computed by considering the average on the firing rates peaks on the recording sites active for that specific phantom movement. Since the data distributions don't meet the assumption of normality required for an ANOVA (analysis of variance), we performed the Friedman test, as a nonparametric test for repeated measured (>2 groups). Additional details about the number of repetitions (n), and p-values for each

experiment are reported in the results and in the corresponding figure legends.

## Reporting summary

Further information on research design is available in the Nature Portfolio Reporting Summary linked to this article.

## Data availability

All data supporting the findings of this study are available within the article and its Supplementary files. Individual de-identified participant data experimental data supporting the findings are immediately and indefinitely available at https://github.com/rossicecilia/intraneural_phantom_leg.git for anyone who wishes to access the data for any purpose. Source data are also provided in this paper. Any additional requests for information can be directed to and will be fulfilled by the corresponding authors. Protocol for human clinical trials is given as part of the reporting summary. Source data are provided with this paper.

## Code availability

Custom code used for analysis is publicly available through https://github.com/rossicecilia/intraneural_phantom_leg.git. Code used for data collection can be made available upon request to the study PIs.

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

## Acknowledgements

The authors are deeply grateful to the three subjects who freely donated months of their life for the advancement of knowledge and for a better future for leg amputees. The funder had no role in the experimental design, analysis, or manuscript preparation or submission. All authors had complete access to the data. All authors authorized the submission of the manuscript, but the final submission decision was made by the corresponding authors. This project has received funding from the European Research Council (ERC) under the European Union's Horizon 2020 research and innovation program (FeelAgain grant agreement no. 759998, S.R.).

## Author contributions

C.R. analyzed the neural data, implemented and tested the SNN-based motor decoders, prepared the figures and wrote the paper; T.S. and P.C. developed the TIME and delivered technical assistance for the human implantation and explanation procedures and reviewed the manuscript; M.B. performed the human surgeries and was responsible for all the clinical aspects of the human study; S.R. designed the study, performed and supervised the human experiments and reviewed the manuscript; E.D., supervised all the analyses, and wrote the paper; G.V. developed the recording software for human experiments, performed the human experiments, supervised all the analyses, prepared the figures and wrote the paper. All authors edited and proofread the manuscript.

## Funding

## Competing interests

G.V. holds shares of "MYNERVA AG", a start-up company dealing with the potential commercialization of noninvasive stimulating wearable for treating neuropathic pain. G.V. serves as a consultant for NeuroOne Medical Technologies Corporation (USA). The other authors do not have anything to disclose.
