## [Transparent Peer Review File · Nature Communications]

Decoding phantom limb movements from intraneural recordings

Corresponding Author: Professor Giacomo Valle

Version 0:

Reviewer comments:

Reviewer #1

(Remarks to the Author)

This paper is globally well written, and the work will be of significance for the field.

Although classification is used in our research on phantom mobility, I am not an expert and will not comment on the technical aspects of the classification itself. Aside from a few specific remarks, my main questions and comments concern how the authors articulate the relationship between their observations (i.e., intraneural signals) and the participants' task execution, as this may have influenced some of their interpretations.

General remarks and questions

- 1) The authors use many abbreviations, which make reading sometimes effortful. Using the full expression instead will make most reading smoother, especially for people that are not experts in the field but still interested.
- 2) As far as I know, the tibial nerve is not involved in extension of the toes and dorsal flexion of the ankle... If so, how can these movements be detected and classified by neural signals measured in this nerve? Therefore, what do the signals related to these movements really represent?
- 3) This work is based on the execution of phantom movements of toes, ankle and knee at a certain rhythm (1/1 s). Yet, (i) many amputees do not have a phantom limb including all articulations (the knee is often absent), and an articulation that is not felt cannot be moved; (ii) even if the articulation is felt, not all amputees can make it move, so not all movements are possible. At most, they try to move it, which results in global contraction of the residual limb instead (e.g., Reilly et al., 2006); (iii) phantom movements are usually much slower than similar intact limb movements. Yet, nothing is reported in this paper on whether the two participants could actually execute the required movements and if the rhythm of 1s –/ 1s was possible for them. The interpretation of the obtained results must take these points into account since the muscle contractions (and thus the motor commands expressed by the intraneural signals) are highly specific for execution of phantom movements (e.g., Jarrassé et al., 2018; Chateaux et al., 2024) but non-specific in cases of “tensing to move” the phantom (Reilly et al., 2006). Moreover, if the participants were not able to follow the rhythm, their movements might have been behind, which could have reduced the classification rate.
- 4) Why using terms such as “attempted movements” and “motor intent” instead of “phantom movement execution”? Movement intentions stay in the central nervous system. The motor activity that the authors measured in the peripheral nerve does not represent intention but are real motor commands. All concerned texts should be rephrased.

Specific remarks and questions

Lines 48-49: The authors state that “limb loss completely disrupts the brain-body communication”. This might be true directly after the amputation but is certainly not true at the long range. Literature shows peripheral sprouting and recapture of the axotomized motor and somatosensory nerves in animals. A recent result of our team suggested similar peripheral reorganisation in human (Chateaux et al., 2024). Therefore, the sensorimotor loop is not disrupted but modified.

Lines 60-62 “However, these solutions [sEMG] lack the specificity and robustness needed for fine-grained, volitional control, particularly when multiple degrees of freedom (DoF) are involved”: Although not yet reported for lower limb amputees, recent

literature on upper limb amputees shows the use of surface EMG associated with phantom movements that might allow fine-grained and volitional control (among others, Jarrassé et al., 2018). It would be preferable to phrase this more cautiously, or at least adding "for lower limb amputees."

Line 70: RPN was already defined before.

Lines 186-189: Why giving both mean and median values? The difference between these clearly show that the data is not normally distributed so the mean and SD values should not be reported.

Lines 191-192: It should be explained how these statistical tests were applied to the data, given that only 2 participants are included. The test is applicable for non-parametric comparison of independent groups and different conditions. What are the independent groups in your model?

Lines 193-194: "the peak firing rate ranges seem to be proportional to the anatomical proximity of the phantom joint...": The reason is not the anatomical proximity but the fact that the involved muscles, targeted by the recorded motor commands, were distally inserted on the foot and proximal on the calf. In other words, these muscles had an action on both the toes and the ankle.

Line 227 Fig. 2B : "Recostruction" should read Reconstruction. What represent "L" and "R" in the figure? Does each "tube" within the nerve represents one neural fibre?

Line 232 Fig. 2D: I suppose that FR on the vertical axes means "Firing Rate" ? This abbreviation should be put in the legend.

Lines 546-547: "We demonstrated to successfully decode attempted movements from the recorded signals...": "Successfully decode" is relative to what was expected. If this method is to be used for real prosthesis control, the successful classification rate must be close to 100%. So, to date, I would resume the results of this study more as a strong prove of concept for using intraneural recording to decode phantom movements.

Reviewer #2

(Remarks to the Author)

Rossi and colleagues show that intrafascicular multichannel electrodes (TIME) implanted into the tibial nerve of two lower limb amputees can be used to detect signals for intended flexion and extension movements of the toe, ankle and knee. They characterise how the spatial distribution of the signals across TIMEs depends on if flexion/extension is attempted, and on which joint is called upon (toe, ankle or knee). Next, they attempt to relate the spatial distribution and ease of TIME activation by an attempted movement to the muscles that are innervated by the tibial nerve, claiming that differences in innervation explain why some attempted movements are easier to detect than others. A spiking neural network is then used on the TIME signals to show that the attempted movements can be decoded, albeit with only modest accuracy. EMG signals are then added to the spiking neural network, which significantly improve accuracy (although to a level equivalent to that with EMG only). Finally, contacts on the TIME are sequentially activated to show that phantom sensations can be evoked. The authors claim that the ability to decode motor intent and stimulate sensations constitutes a bidirectional interface.

The approach of implantable peripheral neural interfacing is certainly exciting, and this work indeed shows that decoding of intended actions (although modest) and sensory stimulation is technically feasible. However, I have concerns over the conceptual flaw of using a decoding system that hopes to be used as a prosthetic for motor control, which requires online, proportional control. There are also methodological limitations such as use of a distance metric I cannot find information about that would need clarification.

Major comments:

1. The authors claim that surface EMG decoders are limited by not accessing appropriate muscles. Whilst true, intramuscular EMG microelectrode arrays have been shown to target deeper muscles with high precision and can be used for interfacing online (see <https://www.medrxiv.org/content/10.1101/2025.07.17.25331429v2> and <https://arxiv.org/abs/2410.10694>). The authors may want to consider discussing these methods in their background. Although surface EMG would indeed does not access the appropriate (especially deep) muscles, their use may be sufficient to capture global activity that could be used for decoding. In fact, the authors show evidence to support this in the current study – a SNN that uses EMG recorded by TIME is as good as EMG+ENG. Given this, I wonder why ENG is needed at all (if EMG has the same accuracy).
2. There is a conceptual flaw to using the global EMG from the thigh for practical applications. Presumably amputees still use their thigh muscles to do other things. How would the decoder differentiate the thigh muscle EMG from this intended thigh activity from the thigh muscle activity generated during attempted knee and ankle movements? Being unable to do so would make the decoder (and hence interface) less stable. This should be discussed.
3. Presumably a prosthetic at the toe, ankle and knee would need proportional control. That is, the user needs to be able to control the amount of flexion/extension at each joint for effective motor control; I struggle to see how a prosthetic would work using an on/off decoding-based approach. Furthermore, co-contraction of muscles around a joint is an essential feature of motor control, allowing for stabilisation of joints – how would this system deal with this motor control problem?
4. I wonder if the experimental design is making differences in flexion and extension-related activity more difficult to

disentangle. If I understand correctly, participants are asked to make a flexion movement for 1 second followed by an extension movement for another 1 second (so alternating flexion then extension). Importantly, there is no return to a resting position between movements (i.e., flexion-rest-extension). If I understand correctly, then participants might not be extending after flexion, they might first relax the flexion (i.e., they are not actively extending but are just letting go of flexion). If that's the case, then some of the activity in the supposed "extension" phase is not actually reflecting active extension. This might also explain why the SNN sometimes mistakes the flexion phase with extension.

5. I do not think that this study shows evidence of a bidirectional interface. There is certainly the potential for one, but the authors have not conducted the study to show this is the case. A suitable proof to make this claim, for example, would be to stimulate during control of a prosthetic to improve behaviour/control.
6. Figure 2 has several issues that need clarifying.
 - a. What do the letters above the plots (EL2 AS-R6) mean?
 - b. I presume the time from 0 to 2 seconds is flexion and then extension, but what is the "start" and "end" in the right panels of 2A?
 - c. The figure legend says that purple shows flexion and brown shows extension but I think this is incorrect in the plot (I may be wrong as I am colourblind)?
 - d. In the figure legend, D) references average firing rates etc. I don't quite understand what this means. Average of what? If the points are outliers, what data goes into the box? I thought there were for TIMEs per subject...
 - e. In plot F, it's not quite clear what the lines actually plot. Are they firing rates from one channel from one TIME? Or average rates across one TIME etc? Also, they need to have plotted the firing rate during rest because it is not 0 (as seen in A). Finally, why does each plot have KNEE, ANKLE and TOE for single and multi-joint?
7. The text on line 263-264 says that muscles supplied by fibres belonging to the tibial nerve were included. However, the tibial nerve does not supply the extensor hallucis longus nor the extensor digitorum longus. This probably impacts the distance metrics and Venn diagrams in Figure 3.
8. I cannot find any information on the "Herdin distance". Consequently, I cannot comment on this whole section and its inferences. I suspect you are trying to make the point that intended movements that involve muscles supplied by the tibial nerve are more detectable on TIME. If so, I don't see why you need a distance measure to make that point.
9. The authors note that the spike rate variability and features of the action potential could be important in differentiating the intended movement. It doesn't seem that these factors have been included in the SVM or MLP but it seems they are features in the SNN. Might it be good to include them as features in the SVM and MLP for a fair comparison?

Minor comments:

1. Please include units for the duration of injury in Table 1.
2. There are several instances where plots/data are reported for S1 only. Can they also be produced for S2, please?
3. The whole section ("The intraneural signal encodes joint-related and direction-related movement of the lower-limb") is quite dense and difficult to follow. It should be shortened to the main points. For example, why are both the mean and median firing rates reported (line 186-189)? Are the mean/median firing rates computed over contacts or direction or something else?
4. I do not think "direction-related" is a faithful representation of what this system achieves. Although perhaps outside the scope of this article, there are other movements around the knee and ankle, like inversion/eversion/rotation that are not accounted for. Instead, the current system decodes flexion and extension only.
5. The authors note that the spatial selectivity/discriminability for different attempted actions depends on where the TIME is implanted. It would be interesting for the authors to note how many degrees-of-freedom can be decoded from a single TIME and if they have any ideas on how to optimise TIME placement to maximise decoding accuracy. It would also be nice to see plots (like in 2E) of ELECTRODE SELECTIVITY x JOINT x DIRECTION. Small note – Figure 2E is not mentioned in the main manuscript.
6. The firing rates for one participant are unusually high. Have the authors removed duplicate action potentials that could be recorded on multiple channels? Could this also be because there are more channels that detect activity for a particular attempted movement (so more spikes detected)?

Reviewer #3

(Remarks to the Author)

Reviewer Report

Manuscript title: Decoding phantom limb movements from intraneural recordings

This manuscript presents a pioneering demonstration of volitional motor decoding from intraneural recordings in lower-limb amputees. Using transverse intrafascicular multichannel electrodes (TIMES) implanted in the tibial nerve, the authors recorded electroencephalographic (ENG) activity during attempted phantom movements (knee, ankle, toes) and decoded motor intent using spiking neural networks (SNNs). The SNN-based decoder significantly outperformed conventional approaches (SVM, MLP) and improved further when incorporating inter-muscular EMG (imEMG) frequency bands. Comparison of motor and sensory maps revealed spatial segregation of afferent and efferent fibers within the tibial nerve. From a surgical and clinical standpoint, this work represents a technically sound and conceptually significant advance toward bidirectional neural interfacing for lower-limb prosthetic control.

Key Results

The study demonstrates that intraneural electrodes can capture multi-unit neural modulation corresponding to joint- and direction-specific phantom limb movements. Neural signals were decoded into six movement classes with accuracies up to ~68% using SNNs. Hybrid decoding (ENG + imEMG bandwidth) further enhanced classification performance, and sensory mapping data show minimal spatial overlap with motor sites, suggesting early segregation of afferent/efferent fibers in the tibial nerve.

Validity

The study is methodologically rigorous, with surgical procedures following accepted microsurgical/ clinical standards (ClinicalTrials.gov NCT03350061). Data acquisition, preprocessing, and analysis are appropriate, (Kruskal–Wallis, ANOVA, t-tests). Although limited by a small sample size (n=2), the study is technically sound and clinically appropriate as a feasibility demonstration.

Level of Support for the Conclusions

The conclusions are well supported by the data, including direction- and joint-specific neural modulation, SNN decoding outperforming traditional models, and consistent sensory–motor segregation. The claims are appropriately framed as feasibility rather than definitive clinical translation.

Significance

This is the first demonstration of direct decoding of lower-limb motor intent from peripheral nerve activity in humans via a 90 days implantable device. The findings have major implications for neuroprosthetic development, surgical targeting strategies and advancing bidirectional neural interfacing for intuitive prosthesis control.

Data and Methodology

Signal acquisition, filtering, and analysis are of higher technical quality. Multi-unit firing analyses are clear and well illustrated.

Analytical Approach

Tissue Response and Signal Selectivity

Signal acquisition, Multi-unit firing, filtering, and analysis are reported (Lines 560–568). Including impedance or long-term stability data would strengthen clinical translation by addressing the targeted chronic implant performance. Please add.

Functional Limitation: Tibial-Nerve-Only Implantation and Missing Peroneal Nerve Activity

As noted in the manuscript (lines 581–585 and Table S2), the study was limited to implantation within the tibial branch of the sciatic nerve. While this approach successfully captures efferent signals associated with plantarflexion and knee flexion, it omits the functional contributions of the common peroneal nerve, which innervates the dorsiflexors and evertors of the leg. From a gait and rehabilitation standpoint, this represents a critical limitation.

The tibial nerve predominantly governs stance-phase mechanics — controlling plantarflexion during push-off and contributing to knee stabilization during loading. In contrast, the peroneal nerve drives swing-phase control, mediating dorsiflexion for foot clearance and eversion for balance. Without peroneal input, decoding is limited to flexor and plantarflexor synergies, leaving the extensor and dorsiflexor commands absent from the neural repertoire.

Clinically, this restricts any future prosthetic implementation to partial gait cycle control, lacking the active dorsiflexion required for swing and the fine balance adjustments needed for stable ambulation. Future designs should therefore include dual-nerve interfacing (tibial + peroneal) to provide comprehensive decoding of gait primitives — stance, propulsion, swing, and landing — for a true bidirectional neuroprosthetic system.

In summary, while tibial nerve implantation validates the feasibility of intraneural motor decoding, inclusion of peroneal nerve activity is essential for achieving complete, physiologically accurate lower-limb prosthetic control.

The authors note (lines 560–568) that the implanted intraneural TIMEs were able to capture both neural (ENG) and inter-muscular EMG (imEMG) signals due to their close placement to residual musculature within the tibial nerve. While this dual signal capture enhances decoding robustness, it introduces a critical surgical trade-off between signal selectivity and long-term tissue response.

From a surgical perspective, penetrating intraneural electrodes like TIMEs provide superior spatial selectivity by accessing individual fascicles. However, this comes with potential biological consequences including local trauma, fibrotic encapsulation, and gradual impedance increase, all of which can compromise chronic recording stability. Although the paper briefly acknowledges possible signal degradation over time, the issue of long-term biocompatibility and encapsulation is central to clinical viability.

Additionally, the presence of transcutaneous leads poses risks of skin irritation, infection, or penetration—especially in transfemoral amputees who bear mechanical load within prosthetic sockets. Future translation will require fully implanted systems with hermetic connectors and flexible cabling to mitigate micromotion and scarring. Periodic impedance monitoring and histological correlation in chronic models will be essential to confirm selectivity retention over time.

In summary, while the acute recordings demonstrate excellent signal selectivity, the long-term balance between nerve penetration depth, tissue response, and mechanical stability will determine the ultimate clinical success of this approach. The analytical framework is robust. The SNN architecture is biologically inspired and computationally efficient, outperforming conventional decoders. The use of neuromorphic methods is well justified and clinically relevant for future real-time applications.

Suggested Improvements

1. Include electrode impedance or signal stability data across the 90-day period.
2. Provide postoperative recovery and safety information.
3. Discuss translation to real-time prosthetic control.
4. Mention any inflammation or electrode migration observed.
5. Expand on future inclusion of peroneal nerve decoding for additional degrees of freedom.

Clarity and Context

The manuscript is clearly written and visually strong. Figures effectively convey anatomical and functional data. The integration of surgical, neurophysiological, and computational perspectives is well balanced.

References

The references are comprehensive, current, and well aligned with the study's interdisciplinary scope.

Expertise and Limitations of Review

This review focuses on surgical and clinical aspects of the study. Computational implementation details were assessed at a conceptual level.

Summary Statement

The manuscript is technically robust and clinically relevant, with strong data supporting feasibility of bidirectional neural control in transfemoral amputees. While limited in cohort size, the findings are compelling and significant.

Overall Assessment

Data quality: Excellent

Analytical robustness: Strong

Support for conclusions: Adequate and convincing

Clinical feasibility: High

Potential significance: Major advance toward bidirectional lower-limb neuroprostheses.

Comments to the Authors

This is an impressive and carefully executed study demonstrating the feasibility of decoding phantom lower-limb movements directly from intraneural recordings. The work bridges advanced neural engineering with clinical surgery and holds strong translational promise for future bidirectional prosthetic systems.

Strengths

The study fills an important translational gap by extending intraneural decoding from the upper to the lower limb, an area historically underexplored in neuroprosthetic research. It demonstrates clear joint- and direction-specific neural modulation, confirming that volitional motor commands can be reliably detected even in the residual peripheral nerves of lower-limb amputees. The authors' use of spiking neural network (SNN)-based decoding, combined with hybrid electroneurographic (ENG) and inter-muscular EMG (imEMG) signal processing, represents a technically innovative and physiologically grounded approach to motor intent recognition. Furthermore, the integration of motor decoding with sensory mapping in the same intraneural interface provides a comprehensive framework for future bidirectional neuroprosthetic systems, merging control and feedback.

Areas for Improvement and Limitations of the study

In addition to the limitations described, several major aspects require explicit discussion:

No Real-Time Control: All analyses were performed offline. Without real-time or closed-loop validation, the robustness of the decoding algorithms under dynamic, real-world conditions remains unknown. This limits the translational applicability of the findings to clinical prosthetic use.

Intraoperative Identification of Nerve Branches: The manuscript does not describe in sufficient detail how the tibial and peroneal branches of the sciatic nerve were distinguished during implantation. Because these branches are in close proximity at that proximal implantation level intraoperative identification is essential to ensure accurate targeting and avoid cross-contamination of signals.

Level of Conclusion: The conclusion (lines 540–570) should be rewritten to clarify that the results demonstrate feasibility of motor decoding, not functional restoration. The claim that the work 'paves the way for bidirectional, neurally controlled prostheses' should be reframed as 'provides preliminary validation of motor decoding feasibility requiring future real-time and chronic testing.'

Implantation and Biocompatibility: While the use of TIMEs avoids the need for bypassing nerves as in TMR, RPNI, or AMI approaches, the requirement for a penetrating intraneural implant still raises issues of fibrotic encapsulation and tissue response. These aspects should be openly discussed as they impact long-term signal stability and selectivity vs. "implant-independent bioboosters" (TMR, RPNI, or AMI).

Figures and Supplementary Material Revisions

- Figure S4: The first graph (S4A) lacks representation of the physiological distribution of motor and sensory fibers in the tibial nerve. A schematic cartography of the 56 electrode sites and their anatomical mapping should be added for clarity.

- Table S2: This should be replaced by a visual figure illustrating the tibial nerve's functional contribution to gait. The figure should show activation phases, primary tibial-innervated muscles, and their biomechanical functions, structured as follows:

Phase / Main Action / Tibial Nerve Muscles /Function /

This visualization would contextualize the tibial nerve's dominant role in stance control and propulsion while underscoring the missing contribution of peroneal activation in swing-phase dorsiflexion and balance.

- Figure S1: A photo of the explanted TIME electrode and skin penetration site should be included. Observations such as local scarring, fibrosis, or skin irritation should be described to document biocompatibility and mechanical tolerance over the implantation period.

Summary of Requested Edits

1. Add explicit note on absence of real-time control.
2. Clarify how tibial and peroneal branches were distinguished intraoperatively.
3. Adjust the conclusion for realistic clinical interpretation.
4. Address chronic biocompatibility and encapsulation observed (after 90 days)
5. Revise Figures S1 and S4 to include implant, physiological, and anatomical mapping details.
6. Replace Table S2 with a schematic figure summarizing tibial nerve function during gait.

These modifications would significantly strengthen the manuscript's clinical relevance, anatomical rigor, and translational transparency.

Overall Impression

This is high-quality translational research combining solid clinical execution with state-of-the-art decoding. Despite the small sample size, the work is significant and persuasive. The manuscript will stand as a benchmark in lower-limb neuroprosthetic interfacing. Congratulations to the authors for a technically ambitious and clinically meaningful contribution.

Version 1:

Reviewer comments:

Reviewer #1

(Remarks to the Author)

Thank you for the modifications in your manuscript following my remarks and questions.
I am satisfied !

Jozina De Graaf

Reviewer #2

(Remarks to the Author)

I am satisfied with the revisions the authors have made in response to my comments.

Reviewer #3

(Remarks to the Author)

The overall revision is satisfactory (and important BIG difference that they stimulate proximal to the bifurcation).

We thank the reviewers for the appreciation of our work and for very important and useful comments that helped us improve the quality of our manuscript. Our point-to-point replies (in blue, together with the modified text in the manuscript, in *italic*) to reviewers' comments (in black) are provided below. We also performed additional checks, corrected the typos and proof-read the manuscript by a native speaker (the most substantial changes are in *light-blue italic*).

REV 1

This paper is globally well written, and the work will be of significance for the field. Although classification is used in our research on phantom mobility, I am not an expert and will not comment on the technical aspects of the classification itself. Aside from a few specific remarks, my main questions and comments concern how the authors articulate the relationship between their observations (i.e., intraneural signals) and the participants' task execution, as this may have influenced some of their interpretations.

We thank the reviewer for highlighting the importance of our study and for all suggestions. In the following letter we address the comments of the reviewer, by point-to-point replies.

General remarks and questions

1) The authors use many abbreviations, which make reading sometimes effortful. Using the full expression instead will make most reading smoother, especially for people that are not experts in the field but still interested.

As suggested, we reduced the number of acronyms and abbreviations used in the paper, in the hope that the reading would be smoother and lighter.

2) As far as I know, the tibial nerve is not involved in extension of the toes and dorsal flexion of the ankle... If so, how can these movements be detected and classified by neural signals measured in this nerve? Therefore, what do the signals related to these movements really represent?

The reviewer raised an important point, since the tibial nerve does not innervate muscles involved in toes and ankle extension. We have two hypotheses on the nature of the neural signal related to extension of those joints:

- First, the implant is above the nerve bifurcation above the knee (still sciatic nerve), and therefore it is not yet divided in peroneal and tibial branches. This is the reason why in the paper we referred to this nerve as the tibial branch of the sciatic nerve. However, the nerve may not have yet completely divided into tibial and peroneal, so some fibers are still present inside, allowing the detection of even extension movements connected to the ankle and toes.

- The second reason relates more to the condition caused by the limb amputation. The agonist and antagonist muscles are normally controlled by common drive (De Luca and Mambrito 1987 J Neurophysiol) and reciprocal inhibition mechanisms (Shindo et al., 1984 Exp Bra Res; Jankowska and Roberts 1972 J Physiol). During extension, the fibers in the muscles are activated and they activate inhibitory spinal interneurons that inhibit the activity of the antagonist (flexor) muscles. In case of an amputated limb, the proprioceptive fibers of the muscles are absent therefore reciprocal inhibition cannot occur. Indeed, no inhibitory modulation has been observed in our recording. This might lead to (at least we cannot exclude) direct activations of both fibers directed to flexors muscles also in case of extension (simultaneous activation of agonist/antagonist). This is also supported by the fact that phantom limb movements are not always specific in amputees (Reilly et al., 2006) as also in other clinical populations (Dietz and Sinkjaer 2007 Lancet Neurol.). This might also suggest that co-contractions (isometric movements) would be difficult to distinguish compared to opposite movements involving the same agonist/antagonist muscle group. Indeed, many of the electrodes showed modulation to both movements (flex/ext).

[REDACTED]

In our study, these hypotheses are also supported by the fact that extension of ankle and toes are in general less decodable compared to the other movements. In other words, as confirmed by the computation of the Herdin's distance the modulation pattern observed in the signals in relation to extension of ankle and toes seems to be more correlated with the percentages and overlapping in the muscle innervated by the tibial branch of the sciatic nerve (flexors) rather than the ones innervated by the peroneal branch of the sciatic nerve (extensors). This could confirm our hypothesis that the observed activation could be related more to inhibitory signals directed to the antagonist flexors muscles, but also that some fibers belonging to the peroneal branch are still present in the implant location, according to the Herdin's distance (higher than comparing flexors muscles and neural activation during extension movements, but still < 0.5) between the neural activation pattern related to the extension of the ankle and toes and the overlapping in the extensors muscles innervated by the peroneal branch of the sciatic nerve.

Since we cannot rule out any of the 2 hypotheses, we decided to comment this and modify as follow:

- 1) we modified Table S2 indicating the muscles innervation.

MUSCLE	KNEE FLEXION	KNEE EXTENSION	ANKLE PLANTARFLEXION	ANKLE DORSIFLEXION	TOES FLEXION	TOES EXTENSION
Gastrocnemius	X		X			
Soleus			X			
Plantaris	X		X			
Popliteus	X					
Tibialis posterior			X			
Flexor hallucis longus			X		X	
Flexor digitorum longus			X		X	
Extensor hallucis longus				X		X
Extensor digitorum longus				X		X
Tibialis Anterior				X		
Peroneus Tertius				X		

Table S2. Muscular innervation of the distal branch of the sciatic nerve. In this table we only included the muscles involved in the movements performed in the task, below the amputation level. The muscles innervated by the deep peroneal branch of the sciatic nerve are indicated in green, while the muscles innervated by the tibial branch of the sciatic nerve are indicated in red. [Rigoard, P. (2020). *Atlas of Anatomy of the Peripheral Nerves: The Nerves of the Limbs-Expert Edition.*; Gray, H. (1878). *Anatomy of the human body (Vol. 8).* Lea & Febiger.

2) We added the Figure S5, where in panel A we represent the different phantom movements asked to the participants.

Figure S5. Types of phantom movements and its muscular representation during gait. A) Flexion and Extension of the knee, Dorsiflexion and Extension of the Ankle. According to common policy, in this study we refer to the dorsiflexion as extension of the ankle, while we use the term flexion of the ankle to describe the plantarflexion. Toes Flexion and Extension. B) ...

3) we change the terminology from ‘tibial branch of the sciatic nerve’ to ‘distal section of the sciatic nerve’

- 4) we re-computed the Venn diagram representing the muscle innervated by the distal branch of the sciatic nerve (and involved in the considered movements), dividing them between extensors (innervated by the peroneal branch of the sciatic nerve) and flexors muscles (innervated by the tibial branch of the sciatic nerve). Also, we re-calculate Herdin distances between neural activity and muscles.

- 5) we modified the text as follow:

In the Discussion, we added the possible 2 hypotheses:

“Interestingly, in our recordings we observed neural activity that could still be associated with these extension movements, albeit less distinct and more difficult to decode compared to other actions. We propose two, non-mutually exclusive, explanations for this finding. First, the recording site was located above the bifurcation of the sciatic nerve, before the separation into the tibial and peroneal branches. At this level, some fibers from both branches are likely intermingled, which could explain why extension-related signals were still observed within a predominantly tibial recording site. Second, the altered neural control mechanisms following amputation may contribute to overlapping activation patterns. In intact physiology, reciprocal inhibition and common drive mechanisms coordinate agonist–antagonist muscle pairs (De Luca and Mambrito, 1987; Shindo et al., 1984; Jankowska and Roberts, 1972). However, in amputees, the absence of proprioceptive feedback may disrupt these inhibitory loops, leading to concurrent activation of both flexor and extensor pathways during attempted extension. This interpretation is consistent with previous evidence of non-specific or co-contracted phantom movements (Reilly et al., 2006; Dietz and Sinkjaer, 2007). Together, these findings suggest that both anatomical factors and altered spinal circuitry after amputation may underlie the reduced separability of neural patterns during extension movements.”

In the Methods, we added info about the electrodes' placements within the nerve:

"... A longitudinal incision was made along the sulcus between the biceps femoris and semitendinosus muscles, positioned at the midpoint of the posterior aspect of the thigh and beginning approximately 4.5 cm proximal to the end of the amputation stump. To expose the sciatic nerve, the semitendinosus muscle was retracted medially while the biceps femoris was moved laterally..."

3) This work is based on the execution of phantom movements of toes, ankle and knee at a certain rhythm (1/1 s). Yet, (i) many amputees do not have a phantom limb including all articulations (the knee is often absent), and an articulation that is not felt cannot be moved; (ii) even if the articulation is felt, not all amputees can make it move, so not all movements are possible. At most, they try to move it, which results in global contraction of the residual limb instead (e.g., Reilly et al., 2006); (iii) phantom movements are usually much slower than similar intact limb movements. Yet, nothing is reported in this paper on whether the two participants could actually execute the required movements and if the rhythm of 1s –/ 1s was possible for them. The interpretation of the obtained results must take these points into account since the muscle contractions (and thus the motor commands expressed by the intraneural signals) are highly specific for execution of phantom movements (e.g., Jarrassé et al., 2018; Chateaux et al., 2024) but non-specific in cases of "tensing to move" the phantom (Reilly et al., 2006). Moreover, if the participants were not able to follow the rhythm, their movements might have been behind, which could have reduced the classification rate.

We agree with the reviewer that this is an important point to highlight. The two participants involved in this study: 1) have vivid perception of their phantom legs, as also shown in a previous study assessing their phantom limb distortions (Petrini et al., 2019 Science Translational Medicine), 2) refer to be able to move their phantom legs (agency), but of course, no residual feedback (neither proprioceptive or visual) is present to control for the effective start and hand of the executed movements (as suggested by the reviewer). For this reason, the muscular and neural contraction patterns might be non-specific causing a decrease in the decoding performance.

To address this point, we add this part and the suggested references in the Discussion:

"Another important consideration concerns the nature and execution of phantom movements in the amputee participants. Both individuals reported vivid phantom limb perception and voluntary control of their phantom legs, as previously described (Petrini et al., 2019). However, in the absence of proprioceptive and visual feedback, the effective timing and specificity of these movements cannot be objectively verified. As a result, the neural and muscular activation patterns may include non-specific contractions of the residual limb rather than articulation-specific motor commands (Reilly et al., 2006; Jarrassé et al., 2018; Chateaux et al., 2024). This lack of sensory feedback may contribute to the variability and reduced separability observed in the decoding of phantom movements, particularly at higher imposed rhythmic rates. These aspects highlight the intrinsic limitations of interpreting motor-related neural signals in the context of deafferented sensorimotor systems."

4) Why using terms such as "attempted movements" and "motor intent" instead of "phantom movement execution"? Movement intentions stay in the central nervous system. The motor activity

that the authors measured in the peripheral nerve does not represent intention but are real motor commands. All concerned texts should be rephrased.

We thank the reviewer for the correction, and we agree that the term “phantom movement execution” would be more appropriate in order to describe the imagination and attempt of moving the phantom limb by the patients, while measured at the peripheral level. Therefore, we changed the occurrences of “motor intent” and “attempted movements” in the paper with “phantom movement execution”.

Specific remarks and questions

Lines 48-49: The authors state that “limb loss completely disrupts the brain-body communication”. This might be true directly after the amputation but is certainly not true at the long range. Literature shows peripheral sprouting and recapture of the axotomized motor and somatosensory nerves in animals. A recent result of our team suggested similar peripheral reorganisation in human (Chateaux et al., 2024). Therefore, the sensorimotor loop is not disrupted but modified.

We modified as suggested:

“.. limb loss *modifies* the brain-body communication”

Lines 60-62 “However, these solutions [sEMG] lack the specificity and robustness needed for fine-grained, volitional control, particularly when multiple degrees of freedom (DoF) are involved”: Although not yet reported for lower limb amputees, recent literature on upper limb amputees shows the use of surface EMG associated with phantom movements that might allow fine-grained and volitional control (among others, Jarrassé et al., 2018). It would be preferable to phrase this more cautiously, or at least adding “for lower limb amputees.

We modified as suggested:

“However, *for lower-limb amputees*, these solutions lack the specificity and robustness needed for fine-grained, volitional control, particularly when multiple degrees of freedom (DoF) are involved”

Line 70: RPN was already defined before.

We fixed this.

Lines 186-189: Why giving both mean and median values? The difference between these clearly show that the data is not normally distributed so the mean and SD values should not be reported.

We apologized for the lack of clarity; as suggested, we removed mean and SD values.

Lines 191-192: It should be explained how these statistical tests were applied to the data, given that only 2 participants are included. The test is applicable for non-parametric comparison of independent groups and different conditions. What are the independent groups in your model?

We thank the review for the comment, in that part of the manuscript we compared the firing rates recorded in the same participant during the different joints movements. We changed the test since

we need a non-parametric test but repeated dependent measures (>2 groups). We use the Friedman test ($p < 0.05$). The results are as it follows:

For S1 (boxplots in **Figure 2D**):

- Ankle and Knee peaks distributions are statistically different with $p = 0.017$;
- Ankle and Toes peaks distributions are statistically different with $p = 0.00001$;
- Knee and Toes peaks distributions are statistically different with $p = 0.00002$;
- Flexion and Extension peaks distributions are not statistically different with $p = 0.48$

While for S2 (boxplots in **Figure S2B**):

- Ankle and Toes peaks distributions are not statistically different with $p = 0.275$;
- Flexion and Extension peaks distributions are statistically different with $p = 0.00002$;

We modified the test as follow:

“Using the non-parametric Friedman test ($p < 0.05$), all the firing rates distributions during ankle, knee and toes movements are pairwise statistically different (Friedman statistical test, $n = 56$, ankle-knee: $p = 0.017$, knee-toes: $p < 0.001$ and ankle-toes: $p < 0.001$).

...

No significant difference emerges in modulation patterns associated with flexion and extension, concerning the same anatomical region (Friedman statistical test, $n=56$, flexion-extension: $p = 0.48$) for S1 (**Figure 2D-A**). Instead, for S2 we observed statistical difference from the firing rates peaks during flexion and extension (Friedman statistical test, $n=56$, flexion-extension: $p = 0.00002$).”

Also, we better cleared how the statistical tests were applied in the corresponding method section:

*“...Additional, statistical tests were applied in order to compare the difference between the distribution of the firing rates (averaged on all the repetitions of a specific phantom movement on each recording site) of the different phantom movements and directions. The distributions represented in the boxplot (**Figure 2D**) are computed by considering the average on the firing rates peaks on the recording sites active for that specific phantom movement. Since the data distributions don't meet the assumption of normality required for an ANOVA (Analysis of Variance), we performed the Friedman test, as a non-parametric test for repeated measured (>2 groups). Additional details about the number of repetitions (n), and p -values for each experiment are reported in the results and in the corresponding figure legends.”*

Lines 193-194: “the peak firing rate ranges seem to be proportional to the anatomical proximity of the phantom joint...”: The reason is not the anatomical proximity but the fact that the involved muscles, targeted by the recorded motor commands, were distally inserted on the foot and proximal on the calf. In other word, these muscles had an action on both the toes and the ankle.

We thank the reviewer for the clarification. To avoid confusion and better explain our point, we corrected our previous statement with the following:

“The peak firing rate ranges seem to be correlated with the insertion level of the muscles targeted by the motor commands of both ankle and toes. In fact, these muscles are active for both types of phantom motion execution but are inserted distally on the foot and proximally on the calf, leading to higher firing rates for the ankle and lower for the toes”.

Line 227 Fig. 2B: “Recostruction” should read Reconstruction.

We fixed the typo.

What represent “L” and “R” in the figure?

They represent Left and Right side of the TIME. We added this information in the captions.

Does each "tube" within the nerve represents one neural fibre?

The ‘tube’ represents the nerve trunk with the fascicles. To improve clarity, we included this information in the caption.

Line 232 Fig. 2D: I suppose that FR on the vertical axes means "Firing Rate"? This abbreviation should be put in the legend.

We fixed this.

Lines 546-547: “We demonstrated to successfully decode attempted movements fr on the recorded signals...”: “Successfully decode” is relative to what was expected. If this method is to be used for real prosthesis control, the successful classification rate must be close to 100%. So, to date, I would resume the results of this study more as a strong prove of concept for using intraneural recording to decode phantom movements.

We thank the reviewer for this suggestion; we modified the sentence

From:

“We demonstrated to successfully decode attempted movements from the recorded signals...”

To:

“We showed a proof of concept for using intraneural recording to decode phantom movement...”

REV 2

Rossi and colleagues show that intrafascicular multichannel electrodes (TIME) implanted into the tibial nerve of two lower limb amputees can be used to detect signals for intended flexion and extension movements of the toe, ankle and knee. They characterise how the spatial distribution of the signals across TIMEs depends on if flexion/extension is attempted, and on which joint is called upon (toe, ankle or knee). Next, they attempt to relate the spatial distribution and ease of TIME activation by an attempted movement to the muscles that are innervated by the tibial nerve, claiming that differences in innervation explain why some attempted movements are easier to detect than others. A spiking neural network is then used on the TIME signals to show that the attempted movements can be decoded, albeit with only modest accuracy. EMG signals are then added to the spiking neural network, which significantly improve accuracy (although to a level equivalent to that with EMG only). Finally, contacts on the TIME are sequentially activated to show that phantom sensations can be evoked. The authors claim that the ability to decode motor intent and stimulate sensations constitutes a bidirectional interface. The approach of implantable peripheral neural interfacing is certainly exciting, and this work indeed shows that decoding of intended actions (although modest) and sensory stimulation is technically feasible. However, I have concerns over the conceptual flaw of using a decoding system that hopes to be used as a prosthetic for motor control, which requires online, proportional control.

We thank the reviewer for highlighting the importance of our study and for all suggestions. In the following part, we address the comments of the reviewer, by point-to-point replies. We highlight that our approach does not want to propose a method for online prosthetic control, but just a proof of concept for using intraneural recording to decode phantom movements, possibly using a single implant for both recording and stimulation. The method is still at its infancy. We modified the manuscript to emphasize this point and avoid confusion.

There are also methodological limitations such as use of a distance metric I cannot find information about that would need clarification.

We apologize for the lack of clarity. We added the description of the Herding distance in the Methods section:

“The Herdin’s Correlation Matrix Distance (CMD) is the distance between two correlation matrices R_1 and R_2 as defined by:

$$d_{corr}(R_1, R_2) = 1 - \frac{\text{tr}\{R_1, R_2\}}{\|R_1\|_f \|R_2\|_f} \in [0,1]$$

Where $\text{tr}\{\cdot\}$ denotes the matrix trace and $\|\cdot\|_f$ the Frobenius norm. It becomes zero if the correlation matrices are equal up to scaling and one if they differ to a maximum extent. For the justification and details see the original paper (Herdin et al., 2005).”.

Major comments:

1. The authors claim that surface EMG decoders are limited by not accessing appropriate muscles. Whilst true, intramuscular EMG microelectrode arrays have been shown to target deeper muscles with high precision and can be used for interfacing online (see <https://www.medrxiv.org/content/10.1101/2025.07.17.25331429v2> and <https://arxiv.org/abs/2410.10694>). The authors may want to consider discussing these methods in their background. Although surface EMG would indeed does not access the appropriate (especially deep) muscles, their use may be sufficient to capture global activity that could be used for decoding. In fact, the authors show evidence to support this in the current study – a SNN that uses EMG recorded by TIME is as good as EMG+ENG. Given this, I wonder why ENG is needed at all (if EMG has the same accuracy).

We thank the reviewer for this important comment. As suggested, in the Introduction we added a part on intramuscular EMG in the introduction highlighting the works mentioned.

“... In some cases, surface electromyographic signals (sEMG) could be used to directly control the prosthetic device (i.e., myoelectric prostheses)⁶. However, these solutions lack the specificity and robustness needed for fine-grained, volitional control, particularly when multiple degrees of freedom (DoF) are involved. A major barrier to the efficacy of sEMG decoders is the challenge of accessing the appropriate muscles. Muscles located deep within the thigh, those that have been anatomically reorganized post-amputation, and those lost due to the amputation present significant challenges for capturing reliable movement-related signals through surface recordings⁷. Notably, intramuscular EMG microelectrode arrays have been shown to target deeper muscles with high precision, and they can be used for interfacing online (Pasquina et al., 2015; Grison et al., 2025; Ferrante et al., 2025). To improve human-machine interfacing, ...”

Regarding the use of ENG, very importantly, the first benefit of having this implant is the fact that the same implantable interfaces could potentially be used for both stimulation and recording, providing direct control and sensory feedback. The implantable system would also avoid the need of electrodes' repositioning, skin-electrode variation, or adaptation of the prosthetic socket to surface electrodes.

In addition, about the combination of ENG+imEMG, we want to highlight that the muscular recording, given the placement of the TIMEs inside the thigh, was inter-muscular. This might allow for a higher quality signal compared to surface, with less cross-talks and variation in skin-electrode impedance. This makes us believe that sEMG would not be sufficient to distinguish among these phantom leg movements. Of course, we did not test it in our study, so dedicated experiments are necessary to test this.

Finally, the performance of the SNN is similar between imEMG and iEMG+ENG for both subjects and the tested movements. So, we agree with the reviewer that, in our scenario, the use of the EMG bandwidth would be enough to have a good motor decoder. However, in principle the ENG would allow to extract more information about missing muscles activation even avoiding involuntary contractions of the residual EMG thigh muscles in a more realistic context of us. In our study, the movements involving intrinsic muscles of lower-leg and foot were limited and also in a passive

scenario (no actual activity with the prosthesis). Experiments involving more movements with even different speeds and forces are required.

We added this part in the Discussion to address these points:

“The combined use of ENG and imEMG provided muscular recordings with higher signal quality than surface EMG, potentially reducing crosstalk and impedance variability. This suggests that surface EMG alone may not be sufficient to distinguish among these phantom leg movements, though this requires dedicated validation. Importantly, these implantable interfaces could serve dual roles for both recording and stimulation, enabling stable, bidirectional control without electrode repositioning. While SNN performance was similar for imEMG and imEMG+ENG, ENG may capture complementary information from muscles no longer active post-amputation, potentially improving decoding.”

2. There is a conceptual flaw to using the global EMG from the thigh for practical applications. Presumably amputees still use their thigh muscles to do other things. How would the decoder differentiate the thigh muscle EMG from this intended thigh activity from the thigh muscle activity generated during attempted knee and ankle movements? Being unable to do so would make the decoder (and hence interface) less stable. This should be discussed.

We thank the reviewer for this comment. In a more realistic context of use, the recorded EMG signals of the residual thigh muscles might potentially contaminate the decodability of phantom movements. For this reason, the use of an ENG filter would allow to remove the unrelated residual muscular activity. However, this is more a speculation given that we do not record data in a more active scenario of use. We decided, therefore, to include this information in the Discussion section:

“In practical applications, residual thigh muscle activity unrelated to phantom movements, recorded by the implants, could interfere with decoding stability. While in the controlled setting this effect was limited, in daily use amputees may recruit the same muscles for other tasks, potentially contaminating the motor decoding. Incorporating ENG-based signals could help disentangle neural from residual muscular activity by filtering out unrelated EMG components. However, future studies in more natural, dynamic scenarios are needed to investigate this effect.”

3. Presumably a prosthetic at the toe, ankle and knee would need proportional control. That is, the user needs to be able to control the amount of flexion/extension at each joint for effective motor control; I struggle to see how a prosthetic would work using an on/off decoding-based approach. Furthermore, co-contraction of muscles around a joint is an essential feature of motor control, allowing for stabilisation of joints – how would this system deal with this motor control problem?

We thank the reviewer for this comment. Even if the purpose of the study is just a proof-of-concept of using intraneural recording to decode phantom movements, we also consider important to define the real-time scenario. The next step of this research will be to investigate if we could decode different speeds and torques related to phantom movements using signals recorded by the TIMEs, as preliminary done in the upper limb (Petrini et al., 2019 Biomed Eng; Cracchiolo et al., 2021 JNE). Then the definition of a real-time decoder has to be tested in human subjects also including

more realistic scenarios where co-contraction for stabilization is present. We add this part in the Discussion:

“An important next step will be to test whether different speeds and torques of phantom movements can be reliably decoded from intraneural signals, building on preliminary findings in the upper limb (Petrini et al., 2019). Realistic prosthetic control also requires handling co-contraction and joint stabilization (Seyedali et al., 2012 BMC). Addressing these factors will be critical for translating intraneural decoding of phantom movements into functional, real-time prosthetic control.”

4. I wonder if the experimental design is making differences in flexion and extension-related activity more difficult to disentangle. If I understand correctly, participants are asked to make a flexion movement for 1 second followed by an extension movement for another 1 second (so alternating flexion then extension). Importantly, there is no return to a resting position between movements (i.e., flexion-rest-extension). If I understand correctly, then participants might not be extending after flexion, they might first relax the flexion (i.e., they are not actively extending but are just letting go of flexion). If that’s the case, then some of the activity in the supposed “extension” phase is not actually reflecting active extension. This might also explain why the SNN sometimes mistakes the flexion phase with extension.

We apologize for the lack of clarity. We do have inter-pauses of 2 seconds between two successive movements. Also, the extension is seen as a return to resting position from the flexion. We acknowledge that during the “extension” phase, some activity may reflect relaxation of the preceding flexion rather than active extension. In future studies, we plan to introduce a brief resting period between movements, which should help disentangle flexion- and extension-related activity more clearly and improve decoding accuracy. We made this clearer in the methods (Experimental Protocol and Movement tasks section):

“The subjects had to move their phantom limbs for each trial as required; one trial lasted 2 active seconds and was followed by 2 seconds of rest (no movement), seen as an inter-pause between two successive repetitions of the same phantom movement. The total movement consists of 1 second for the flexion, immediately followed by 1 second for extension.”

5. I do not think that this study shows evidence of a bidirectional interface. There is certainly the potential for one, but the authors have not conducted the study to show this is the case. A suitable proof to make this claim, for example, would be to stimulate during control of a prosthetic to improve behaviour/control.

We agree with the reviewer. We removed the statement “By using dataset recorded from TIMEs and integrating it with SNN-based decoding, and sensory mapping, our work demonstrates a complete bidirectional intraneural interface for lower-limb neuroprosthetics” (line 119) from the manuscript. Also, we corrected the statements claiming that our approach paved the way to bidirectional interfaces/prosthetics with the following rephrasing: “provides preliminary validation of motor decoding feasibility for bidirectional, neurally-controlled prosthetic limbs”. We kept the term bidirectional (or bidirectionally), to refer to the use of the TIME implants as both a recording and stimulation device (even if the two steps were carried out in separate experiments).

6. Figure 2 has several issues that need clarifying.

a. What do the letters above the plots (EL2 AS-R6) mean?

It means Electrode 2 (TIME #2) active site R6 (right side #6). We included this info in the caption.

b. I presume the time from 0 to 2 seconds is flexion and then extension, but what is the “start” and “end” in the right panels of 2A?

It is the 1 sec of extension and the 1-sec of flexion in the same plot. We added this info.

c. The figure legend says that purple shows flexion and brown shows extension, but I think this is incorrect in the plot (I may be wrong as I am colourblind)?

In both the caption and Fig.2 we show flexion in brown and extension in purple, as correctly indicated by the legend of Fig. 2A. We corrected the typo error in the legend of Fig. 2B.

d. In the figure legend, D) references average firing rates etc. I don't quite understand what this means. Average of what? If the points are outliers, what data goes into the box? I thought there were for TIMEs per subject...

We apologize for the confusion. We rephrased the caption:

“Distribution of the average firing rates computed on all repetitions for each joint (right) and movement direction (left), across the recording sites revealing neural modulation. On each box, the central mark ... “

e. In plot F, it's not quite clear what the lines actually plot. Are they firing rates from one channel from one TIME? Or average rates across one TIME etc? Also, they need to have plotted the firing rate during rest because it is not 0 (as seen in A). Finally, why does each plot have KNEE, ANKLE and TOE for single and multi-joint?

Yes, they are example of individual channels from one specific site on a TIME electrode, averaged across repetitions. KNEE, ANKLE, TOES refer to the conditions so are present in each plot in panel F, while the plot titles (Single- or Multi-joint) refer to the joint selectivity observed for that specific channel. Single-joint indicates a channel modulating ONLY with a specific joint movement; while multi-joint a channel modulating for more than 1 joint.

To improve clarity, we added more info in the caption:

“Multi-unit activity on three different, active sites showing significant single-joint modulation only for a single motion type (single-joint). A channel showing multi-joint selectivity, so neural modulation for more than one joint, is also displayed (right). Different color lines represent the average signal across all repetition for a specific movement: knee (orange), ankle (red) or toes (light blue) movements. Mean and SEM are reported.”

Nevertheless, as also addressed in the manuscript, we observed a relevant variability in the interactions rest intervals recorded in between different movements type (ankle, toes, knee). We were afraid that adding the overlap of rest would create further confusion (too many lines), making it difficult to appreciate the presence of selective activation in different movement types.

7. The text on line 263-264 says that muscles supplied by fibres belonging to the tibial nerve were included. However, the tibial nerve does not supply the extensor hallucis longus nor the extensor digitorum longus. This probably impacts the distance metrics and Venn diagrams in Figure 3.

We thank the reviewer for this comment. Firstly, we modified the text from:

“We only included muscles innervated by fibers belonging to the tibial nerve³³, where the implants were located, as listed in **Table S2**.”

To:

“We included muscles innervated by fibers belonging to the distal branch of the sciatic nerve and involved in the performed movements (list in **Table S2**).”

We modified Table S2 indicating the muscles innervation including more details in the caption as well (peroneal/tibial).

MUSCLE	KNEE FLEXION	KNEE EXTENSION	ANKLE PLANTARFLEXION	ANKLE DORSIFLEXION	TOES FLEXION	TOES EXTENSION
Gastrocnemius	X		X			
Soleus			X			
Plantaris	X		X			
Popliteus	X					
Tibialis posterior			X			
Flexor hallucis longus			X		X	
Flexor digitorum longus			X		X	
Extensor hallucis longus				X		X
Extensor digitorum longus				X		X
Tibialis Anterior				X		
Peroneus Tertius				X		

Table S2. Muscular innervation of the distal branch of the sciatic nerve. In this table we only included the muscles involved in the movements performed in the task, below the amputation level. The muscles innervated by the deep peroneal branch of the sciatic nerve are indicated in green, while the muscles innervated by the tibial branch of the sciatic nerve are indicated in red. [Rigoard, P. (2020). *Atlas of Anatomy of the Peripheral Nerves: The Nerves of the Limbs-Expert Edition*.; Gray, H. (1878). *Anatomy of the human body (Vol. 8)*. Lea & Febiger.

We re-computed the Venn diagram representing the muscle innervated by the distal branch of the sciatic nerve (and involved in the considered movements), dividing them between extensors (innervated by the peroneal branch of the sciatic nerve) and flexors muscles (innervated by the

tibial branch of the sciatic nerve). Also, we re-calculate Herdin distances between neural activity and muscles.

8. I cannot find any information on the “Herdin distance”. Consequently, I cannot comment on this whole section and its inferences. I suspect you are trying to make the point that intended movements that involve muscles supplied by the tibial nerve are more detectable on TIME. If so, I don’t see why you need a distance measure to make that point.

We apologize for the lack of clarity. We added the description of the Herdin’s distance in the Methods section and cited the original paper proposing this metric as a measure of the correlation between matrices:

“The Herdin’s Correlation Matrix Distance (CMD) is the distance between two correlation matrices R_1 and R_2 as defined by:

$$d_{corr}(R_1, R_2) = 1 - \frac{tr\{R_1, R_2\}}{\|R_1\|_f \|R_2\|_f} \in [0,1]$$

Where $tr\{\cdot\}$ denotes the matrix trace and $\|\cdot\|_f$ the Frobenius norm. It becomes zero if the correlation matrices are equal up to scaling and one if they differ to a maximum extent. For the justification and details see the original paper (Herdin et al., 2005).”.

The reason of including such a distance was mostly to give a numerical quantification on the fact that muscular innervation, indeed, predict very well the goodness of movement decodability since

the recording of muscles supplied by the nerve are better detectable. To clarify, we added this sentence in the Results section:

“This metric was exploited in order to provide a numerical quantification of the fact that muscular innervation can indeed predict the quality in the decodability of the various movements. The results indicated ...”

9. The authors note that the spike rate variability and features of the action potential could be important in differentiating the intended movement. It doesn't seem that these factors have been included in the SVM or MLP but it seems they are features in the SNN. Might it be good to include them as features in the SVM and MLP for a fair comparison?

We apologize for the lack of clarity. In **Fig S3**, we provided the spiking-based feature (spike rate) also for SVM and MLP for both subjects, showing low decoders accuracy. The spike rate, as the sum of the number of spikes (obtained by applying a thresholding method (as illustrated in the methods) in temporal windows, can be seen as a representation of how the number of spikes (e.g. the firing rate) changes across time in the data sample (window of 150 ms). Therefore, by considering data samples containing the evolution of the FR in a certain time window, we can state that the FR variability is already accounted for.

We modified the text to highlight it more:

“Both SVM and MLP were firstly tested using variety of signal features (*root mean square (RMS) and power*), including also spiking-based features, (such as spike rate), (**Supplementary Figure 3**) to find the best possible. ... “

Minor comments:

1. Please include units for the duration of injury in Table 1.

As suggested, we included this information in Table 1.

2. There are several instances where plots/data are reported for S1 only. Can they also be produced for S2, please?

We reported S2 data in main Figure 4 and 5 and all the rest of analyses in Supplementary Figures:

Fig. S2

Fig S2. Neural response during phantom movements in S2. A) Percentage of significant modulation among all 56 channels. Data for silent vs active, flexion vs extension and knee vs ankle vs toes are reported. n indicates the sample size. B) Average firing rates across all channels for each direction and joint involved. On each box, the central mark indicates the median, and the bars the interquartile ranges, IQR. Points indicate the outliers. C) Each of the 4 TIME are reported for the 3 joint movements, showing the significant modulation for each individual channel. A channel colored in purple shows only significant modulation for flexion, in brown only for extension, both colors both directions and white no modulation. * $p < 0.05$. Data from S2.

Fig. S4

Fig S4. Motor modulation and sensory restoration in S2. A) Distribution of the electrodes showing significant motor modulation compared to those evoking sensation in the ankle or foot areas. B) Each of the 4 TIME are reported for the 2 joints, showing the significant motor modulation (blue) or sensory evoked response (green) for each individual channel. A channel colored in blue shows only significant motor modulation, in green only for evoked-sensation, both colors both responses and white no response. C) Modulation matrices showing the percentage of electrodes significantly modulating only during movement, evoking only sensations or both for the two joints. N=56. Data from S2.

3. The whole section (“The intraneural signal encodes joint-related and direction-related movement of the lower-limb”) is quite dense and difficult to follow. It should be shortened to the main points. For example, why are both the mean and median firing rates reported (line 186-189)? Are the mean/median firing rates computed over contacts or direction or something else?

We apologize for the lack of clarity. We modified the section. In particular, we removed mean and SD values. The medians are calculated over electrode contacts. We added this info in the text.

“The median firing rates, computed over electrode contacts, were 24.81 Hz [IQR, 30.62] for the knee, ...”

4. I do not think “direction-related” is a faithful representation of what this system achieves. Although perhaps outside the scope of this article, there are other movements around the knee and ankle, like inversion/eversion/rotation that are not accounted for. Instead, the current system decodes flexion and extension only.

We agree with the reviewer. As suggested, we removed the “direction-related” definition, substituting it with flexion/extension-related both in the results and discussion.

5. The authors note that the spatial selectivity/discriminability for different attempted actions depends on where the TIME is implanted. It would be interesting for the authors to note how many degrees-of-freedom can be decoded from a single TIME and if they have any ideas on how to optimise TIME placement to maximise decoding accuracy.

We thank the reviewer for this suggestion. We added in the text and in a supplementary figure (Fig. S6) the decodability for individual TIME:

Figure S6. Decoder performance for individual and multiple TIMEs. A) Schematic of the 4 TIME electrodes and relative active sites. B) Top: Spiking Neural Network decoder performance on individual TIMEs, averaged across 5 folds. Bottom: Decoder Performance according to increasing number of TIMEs. Averaged mean test accuracy (on 5-fold cross validation) on the 4 TIME electrodes (for the single electrode performance), or different combinations of TIMEs (for 2 and 3 electrode performance).

In the Method section, we added a paragraph on the individual and multiple TIMEs:

“Single and multi-electrode contribution in SNN decoding

In order to investigate how the amount of information recorded by each TIME was involved in the overall decodability of the attempted movements, we also evaluated a variation of the SNN model, which used a reduced number of input neurons (from 56 to 14), separately on each TIME. We used a 5-Fold cross validation approach, reporting the averaged test accuracies across the different folds (mean \pm standard deviation) in Figure S6B, for both S1 and S2. The performance on different TIME electrodes, on the same data split, were compared using t-paired test.

Additionally, in order to assess how the decodability was influenced by the number of electrodes, we performed analogous analysis using increasing number of TIMEs. We trained a SNN with 28 and 32 input neurons, respectively, on all the possible combinations of 2 and 3 TIMEs. For each combination, we computed the average test performance using 5-fold cross validation. Finally, the accuracies were also averaged across all the combinations and reported in Figure S6C, comparing them with the average across single TIMEs as well as the accuracy achieved using all 4 TIMEs. We statistically

compared performances, and we calculated Pearson's correlation coefficient (R) to quantify the effect of adding multiple electrodes."

In the Results section:

"We also, considered how the spatial selectivity and movement discriminability might depend on how many TIMEs are implanted into the targeted nerve. To this aim, we calculated decoder performance using individual or multiple TIMEs (**Figure S6**). In both S1 and S2, we can appreciate how 2 of the 4 TIMEs appear to be more selective than the others (t -paired test, $p < 0.05$). Also, it's evident how increasing the number of electrodes greatly improves the overall decodability of the attempted actions ($R = 0.93$ for S1; $R = 0.86$ for S2), going from an average classification accuracy of 34% in S1 and 44% in S2 using a single electrode, up to 56% in S1 and 68% in S2 using all four electrodes (**Figure S6B**)."

At the moment, we cannot predict the exact TIME placement relative to the motor fibers. However, modelling studies (Raspovic et al., 2017 Proc. IEEE and Zelenowski et al., 2020 JNER) showed the possibility to predict the optimal number of TIME to be implanted to cover the cross-section of a target nerve. We added this information in the methods:

"The exact placement of the TIMEs relative to motor fibers is unfortunately undetermined. However, modeling studies (Raspovic et al., 2017 Proc IEEE; Zelenowski et al., 2020 JNER) suggest it is possible to estimate the optimal number of TIMEs required to cover the cross-section of a target nerve"

It would also be nice to see plots (like in 2E) of ELECTRODE SELECTIVITY x JOINT x DIRECTION.

We thank the reviewer for the comment. We understand that a single, unique figure containing all the overlapping information would give a final and comprehensive visual report. Nevertheless, all the information can be extracted by the overlapping between the three subfigures in Figure 3B, representing the ELECTRODE SELECTIVITY X DIRECTION separately for each JOINT. We think that all the information together in a single plot would result in a redundant display of information (since there is also a limited number of figures in the journal formatting). It would also require the use of 6 different colors (3 joints x 2 directions) making it difficult to read. By representing separately, the information for each joint, we hoped to give a clearer representation of the modulation pattern observed.

Small note – Figure 2E is not mentioned in the main manuscript.

Fixed. It is also mentioned in line 286.

6. The firing rates for one participant are unusually high. Have the authors removed duplicate action potentials that could be recorded on multiple channels? Could this also be because there are more channels that detect activity for a particular attempted movement (so more spikes detected)?

We thank the reviewer for this comment. We did not perform spike sorting with removal of spike duplication. Since we agree that more channels could have recorded the same activity for the same attempted movement and thus more spikes are detected, we added this point in the limitation section:

“Future studies should consider implementing spike sorting or duplication removal to more accurately estimate firing rates and account for multi-channel detection of the same units.”

REV 3

This is an impressive and carefully executed study demonstrating the feasibility of decoding phantom lower-limb movements directly from intraneural recordings. The work bridges advanced neural engineering with clinical surgery and holds strong translational promise for future bidirectional prosthetic systems.

We thank the reviewer for his/her introductory comment highlighting the potential translational impact of our study.

Strengths

The study fills an important translational gap by extending intraneural decoding from the upper to the lower limb, an area historically underexplored in neuroprosthetic research. It demonstrates clear joint- and direction-specific neural modulation, confirming that volitional motor commands can be reliably detected even in the residual peripheral nerves of lower-limb amputees. The authors' use of spiking neural network (SNN)-based decoding, combined with hybrid electroneurographic (ENG) and inter-muscular EMG (imEMG) signal processing, represents a technically innovative and physiologically grounded approach to motor intent recognition. Furthermore, the integration of motor decoding with sensory mapping in the same intraneural interface provides a comprehensive framework for future bidirectional neuroprosthetic systems, merging control and feedback.

We agree with the strengths listed by the reviewer, in particular about the technological innovation.

Areas for Improvement and Limitations of the study. In addition to the limitations described, several major aspects require explicit discussion:

We thank the reviewer for all the suggestions. In the following letter we address the comments, by point-to-point replies.

No Real-Time Control: all analyses were performed offline. Without real-time or closed-loop validation, the robustness of the decoding algorithms under dynamic, real-world conditions remains unknown. This limits the translational applicability of the findings to clinical prosthetic use.

We agree with this comment. We added more about this point in the limitation section:

“Finally, although we demonstrated motor decoding, this was done in an offline environment. This is only a proof-of-concept of using intraneural signals to decode phantom movements. Real-time use would introduce more variability and noise, which could challenge the decoder's performance. The robustness of the decoding algorithms under dynamic, real-world conditions remains unknown. Moreover, the use of the same electrodes also for delivering neurostimulation to restore sensory feedback would introduce stimulation-artifacts. Techniques based on blanking⁵⁴ or time-division multiplexing⁵⁵ should be implemented to allow a stable bidirectional configuration. Future work should explore how the neural decoding holds up under more dynamic and bidirectional conditions.”

Intraoperative Identification of Nerve Branches: The manuscript does not describe in sufficient detail how the tibial and peroneal branches of the sciatic nerve were distinguished during implantation.

Because these branches are in close proximity at that proximal implantation level intraoperative identification is essential to ensure accurate targeting and avoid cross-contamination of signals.

We apologize for the lack of clarity. We added more information about the surgical procedure in the Methods ('Participant recruitment and surgical procedure'):

From:

"Four TIMEs⁵⁶ (14 active sites each) were obliquely implanted in the tibial branch of the sciatic nerve of each subject. The surgical approach used to implant TIMEs has been extensively reported elsewhere⁵⁷. Briefly, under general anesthesia, through a skin incision over the sulcus between the biceps femoris and semitendinosus muscles, the tibial nerve was exposed to implant 4 TIMEs. A segment of the microelectrodes cables was drawn through 4 small skin incisions 3 to 5 cm higher than the pelvis ilium. The cable segments were externalized (and secured with sutures) to be available for the transcutaneous connection with a neural stimulator. After 90 days, the microelectrodes were removed under an operating microscope in accordance with the protocol and the obtained permissions."

To:

"The electrode implantation procedures were conducted under general anesthesia. A longitudinal incision was made along the sulcus between the biceps femoris and semitendinosus muscles, positioned at the midpoint of the posterior aspect of the thigh and beginning approximately 4.5 cm proximal to the end of the amputation stump. To expose the sciatic nerve, the semitendinosus muscle was retracted medially while the biceps femoris was moved laterally. Each participant received 4 TIME-4H electrodes.

For each electrode, the surgeon created a small window in the epineurium, allowing transverse passage through the visible fascicles. The electrode was then drawn through the nerve so that its active (stimulating) sites made direct contact with the fascicular tissue. Once positioned, the electrode was secured to the epineurium using sutures through the fixation tabs. After all electrodes were implanted, a fascial flap was raised by cutting and folding a patch of fascia around the electrode cables, which was then sutured to the underlying tissue for stabilization. The electrode leads were tunneled subcutaneously through the thigh and exteriorized via small incisions made on the anterolateral aspect of the thigh, a few centimeters below the iliac crest, enabling transcutaneous connection to an external neurostimulator.

During surgery, electrode impedance was continuously monitored using the stimulator's built-in impedance check function to ensure proper contact and functionality. Following implantation, each active site was tested, and only contacts with impedances below 100 k Ω were considered functional, confirming their capacity to deliver current to the nerve. Each surgery lasted approximately four hours. At the conclusion of the study, all implanted electrodes were surgically explanted from both participants."

It is important to note that, although sensations were evoked only within the tibial nerve territory (see also Petrini et al., 2019, *Science Translational Medicine*), the implantation was performed proximal to the knee bifurcation, where the sciatic nerve may not yet be fully separated into its

tibial and peroneal branches. As a result, some fibers of extensor muscles could still be present within the nerve trunk, potentially explaining the detection of extension-related movements involving the ankle and toes (hypothesis).

Level of Conclusion: The conclusion (lines 540–570) should be rewritten to clarify that the results demonstrate feasibility of motor decoding, not functional restoration. The claim that the work 'paves the way for bidirectional, neurally controlled prostheses' should be reframed as 'provides preliminary validation of motor decoding feasibility requiring future real-time and chronic testing.'

Following reviewer's suggestion, we rewrote the conclusion:

From:

“Overall, the use of a compact, biologically inspired SNN decoder enabled accurate and efficient decoding of volitional lower-limb motor intent from intraneural signals, highlighting the potential of neuromorphic approaches in developing next-generation bidirectional neuroprostheses.”

To:

“Overall, the use of a compact, biologically inspired SNN decoder enabled accurate and efficient decoding of volitional lower-limb motor intent from intraneural signals, providing a preliminary validation of neuromorphic approaches for motor decoding. These approaches require future real-time and chronic testing.”

In the abstract and introduction, we replaced 'paves the way for bidirectional, neurally controlled prostheses' with *'provides preliminary validation of motor decoding feasibility'*.

Implantation and Biocompatibility: While the use of TIMEs avoids the need for bypassing nerves as in TMR, RPNI, or AMI approaches, the requirement for a penetrating intraneural implant still raises issues of fibrotic encapsulation and tissue response. These aspects should be openly discussed as they impact long-term signal stability and selectivity vs. “implant-independent bioboosters” (TMR, RPNI, or AMI).

We thank the reviewer for this comment. We added this part in the Discussion ('Limitation of the study'):

From:

“We cannot extrapolate from these findings what would occur over longer periods of time when trying to decode phantom movements. Over time, signal quality can decline due to factors like tissue response around the electrode, slight shifts in position, or nerve adaptation, as already shown for TIME in humans^{52,53}. Studies specifically devoted to this issue are required and are well beyond the scope of the experiments reported here.”

To:

“From these findings we cannot predict long-term outcomes when decoding phantom movements. Signal stability may deteriorate over extended periods due to factors such as tissue responses surrounding the electrode, minor positional shifts, or nerve adaptation, as previously reported for TIME implants

in humans^{52,53}. Importantly, while the use of TIMEs eliminates the need for nerve rerouting procedures such as TMR, RPNI, or AMI, their penetrating intraneural nature introduces challenges related to fibrotic encapsulation and chronic tissue response. These biocompatibility aspects critically influence long-term signal reliability and selectivity and warrant careful consideration when comparing TIMEs to implant-independent approaches. Dedicated longitudinal studies are therefore required to fully assess chronic stability and functional durability.”

Figures and Supplementary Material Revisions

- Figure S4: The first graph (S4A) lacks representation of the physiological distribution of motor and sensory fibers in the tibial nerve. A schematic cartography of the 56 electrode sites and their anatomical mapping should be added for clarity.

We thank the reviewer for the comment, we have reported in Figure 1B the motor and sensory innervation of the targeted nerve. Unfortunately, we cannot infer the exact location of the electrodes within the nerve, so we cannot provide an accurate anatomical mapping. Nevertheless, we have reported in Valle et al 2022 Biomaterials an in-silico re-construction of the nerve and the electrode location (of S2) based on the surgical procedure and the histology (part in Fig. 2B). We also showed more detailed analysis on the nerves after explantation.

(b) 3D realistic reconstruction of the patient nerve using hybrid modeling. A geometrical reconstruction of a curving fascicle and the complete nerve representation based on the segmented cross-sections of the nerve with an implanted electrode. In the top right of the panel, a detailed SEM image of a stimulation AS with its dimensions is shown.

- Table S2: This should be replaced by a visual figure illustrating the tibial nerve's functional contribution to gait. The figure should show activation phases, primary tibial-innervated muscles, and their biomechanical functions, structured as follows: Phase / Main Action / Tibial Nerve Muscles /Function / This visualization would contextualize the tibial nerve's dominant role in stance control and propulsion while underscoring the missing contribution of peroneal activation in swing-phase dorsiflexion and balance.

As suggested, we modified Table S2:

MUSCLE	KNEE FLEXION	KNEE EXTENSION	ANKLE PLANTARFLEXION	ANKLE DORSIFLEXION	TOES FLEXION	TOES EXTENSION
Gastrocnemius	X		X			
Soleus			X			
Plantaris	X		X			
Popliteus	X					
Tibialis posterior			X			
Flexor hallucis longus			X		X	
Flexor digitorum longus			X		X	
Extensor hallucis longus				X		X
Extensor digitorum longus				X		X
Tibialis Anterior				X		
Peroneus Tertius				X		

Table S2. Muscular innervation of the distal branch of the sciatic nerve. In this table we only included the muscles involved in the movements performed in the task, below the amputation level. The muscles innervated by the deep peroneal branch of the sciatic nerve are indicated in green, while the muscles innervated by the tibial branch of the sciatic nerve are indicated in red. [Rigoard, P. (2020). *Atlas of Anatomy of the Peripheral Nerves: The Nerves of the Limbs-Expert Edition.*; Gray, H. (1878). *Anatomy of the human body (Vol. 8).* Lea & Febiger.

We added the Figure S5 representing: A) the different phantom movements asked to participants in this study, B) The involvement of the muscles (and the relative nerve) in the different gait phases:

Figure S5. Types of phantom movements and its muscular representation during gait. A) Flexion and Extension of the knee, Dorsiflexion and Extension of the Ankle. According to common policy, in this study we refer to the dorsiflexion as extension of the ankle, while we use the term flexion of the ankle to describe the plantarflexion. Toes Flexion and Extension. B) Representation of muscle activation during a gait cycle. The color bars indicate periods where the muscles are active during the gait cycle. Phase, function and nerve innervation are displayed.

- Figure S1: A photo of the explanted TIME electrode and skin penetration site should be included. Observations such as local scarring, fibrosis, or skin irritation should be described to document biocompatibility and mechanical tolerance over the implantation period.

Following reviewer's suggestion, we added pictures (percutaneous cables, fibrosis) in Figure S1.

Fig S1. Surgical implantation of the intraneural electrodes in the tibial nerve. A-B) The electrodes are positioned to cross transversally the amputated tibial branch of the sciatic nerve. C) The electrode cables are tunneled through the thigh and pulled out of the leg through small incisions just a few centimeters below the iliac crest, to enable transcutaneous connection with the neurorecorder. D-E) The placement of the implants within the thigh is shown in the X-ray pictures. Adapted from Petrini et al.²¹. F-G) Left: Hematoxylin and Eosin (HE) staining of implanted nerve sections in S2 is displayed. Arrows indicate the electrode. Right: Photomicrographs of HE stained section of the tibial nerve with an electrode implanted at 20x magnification showing the electrode surrounded by a connective tissue capsule and aggregates of macrophages. Fibrotic encapsulation is indicated within the dashed area (after 90 days). Images taken from Valle et al.

In addition, since biocompatibility is not the aim of our study, we now refer to our study where we provide results about tissue-electrode reactions at the nerve level during this trial. Indeed, more information about biological reactions during the implant can be found in Valle et al., 2022 Biomaterials. We added this information in the Methods ('Participant recruitment and surgical procedure'):

"... Detailed characterization of tissue–electrode interactions and biological responses at the nerve level following implantation of TIME electrodes have been reported in a previous work (Valle et al., 2022, Biomaterials)."

We also included more information about skin penetration, biocompatibility and mechanical tolerance over the implantation period in the Methods:

"The electrode implantation procedures were conducted under general anesthesia. A longitudinal incision was made along the sulcus between the biceps femoris and semitendinosus muscles, positioned at the midpoint of the posterior aspect of the thigh and beginning approximately 4.5 cm proximal to the end of the amputation stump. To expose the sciatic nerve, the semitendinosus muscle was retracted medially while the biceps femoris was moved laterally. Each participant received 4 TIME-4H electrodes.

For each electrode, the surgeon created a small window in the epineurium, allowing transverse passage through the visible fascicles. The electrode was then drawn through the nerve so that its active (stimulating) sites made direct contact with the fascicular tissue. Once positioned, the electrode was secured to the epineurium using sutures through the fixation tabs. After all electrodes were implanted, a fascial flap was raised by cutting and folding a patch of fascia around the electrode cables, which was then sutured to the underlying tissue for stabilization. The electrode leads were tunneled subcutaneously through the thigh and exteriorized via small incisions made on the anterolateral aspect of the thigh, a few centimeters below the iliac crest, enabling transcutaneous connection to an external neurostimulator.

...

The histological analysis performed after explantation (90 days) revealed that the intraneural electrodes generated a Foreign Body Response (FBR) (Figure S1G). A significant FBR was only present at the implantation level, with the proximal and distal part of the nerve remaining unaffected given the low number of macrophages. No severe adverse events (e.g., infections) have been observed during the trial, while perceptual thresholds, sensation location changes and some electrode failures have been reported (Valle et al., 2022 Biomaterials)."

Overall Impression: This is high-quality translational research combining solid clinical execution with state-of-the-art decoding. Despite the small sample size, the work is significant and persuasive. The manuscript will stand as a benchmark in lower-limb neuroprosthetic interfacing. Congratulations to the authors for a technically ambitious and clinically meaningful contribution.

We sincerely thank the reviewer for his/her positive and encouraging feedback. We greatly appreciate the recognition of our work as a meaningful translational effort combining clinical

execution with advanced decoding methods. It is gratifying to know that the study is viewed as a valuable contribution and potential benchmark in lower limb neuroprosthetic interfacing.